



# Evaluation of isoprene emissions from the coupled model SURFEX-MEGANv2.1

Safae Oumami[1], Joaquim Arteta[1], Vincent Guidard[1], Pierre Tulet[2], and Paul D. Hamer[3]

[1]CNRM, Université de Toulouse, Météo-France, CNRS, Toulouse, France
[2]Laboratoire d'Aérologie, Université Paul Sabatier, CNRS, Toulouse, France
[3]NILU, Kjeller, Norway

**Correspondence:** Safae Oumami (safae.oumami@meteo.fr)

**Abstract.** Isoprene, a key biogenic volatile organic compound, plays a pivotal role in atmospheric chemistry. Due to its high reactivity, this compound contributes significantly to the production of tropospheric ozone in polluted areas, and to the formation of secondary organic aerosols.

The assessment of biogenic emissions is of great importance for regional and global air quality evaluation. In this study, we

have implemented the biogenic emissions model MEGANv2.1 (Model of Emission of Gases and Aerosols from Nature, version 2.1) in the surface model SURFEXv8.1 (SURface EXternalisée in french, version 8.1). This coupling aims to improve the estimation of biogenic emissions using the detailed vegetation type-dependent treatment included in the SURFEX vegetation ISBA scheme. This scheme provides to MEGAN vegetation-dependent parameters allowing a more precise estimation of biogenic fluxes (e.g., leaf area index, soil moisture, wilting point data).

The present study focuses on the assessment of the SURFEX-MEGAN model isoprene emissions. The evaluation of the coupled SURFEX-MEGAN model results was carried out by conducting a global isoprene emissions simulation in 2019 and comparing the simulation results with other MEGAN-based isoprene inventories. The coupled model estimates a total global isoprene emission of 442Tg in 2019. The estimated isoprene is within the range of results obtained with other MEGAN-based isoprene inventories, ranging from 311Tg to 637Tg. The spatial distribution of SURFEX-MEGAN isoprene is consistent with

other studies, with some differences located in low isoprene emission regions.

Several sensitivity tests were conducted to quantify the impact of different model inputs and configurations on isoprene emissions. Using different meteorological forcings resulted in a +/-5% change in isoprene emission using MERRA and IFS, respectively, compared with ERA5. The impact of using different emission factors data was also investigated. The use of PFT spatial coverage and PFT-dependent emission potential data resulted in a 14% reduction compared to using the isoprene emission po-

tential gridded map. A significant reduction of around 38% in global isoprene emissions, was observed in the third sensitivity analysis, which applied a parameterization of soil moisture deficit, particularly in certain regions of Australia, Africa and South America.

The significance of coupling the SURFEX and MEGAN models lies particularly in the ability of the coupled model to be able to be forced with meteorological data from any time period. This means, for instance, that this system can be used to predict



biogenic emissions in the future. This aspect of this work is significant given the changes that biogenic organic compounds are expected to undergo as a result of changes in their climatic factors.

# 1 Introduction

Volatile Organic Compounds (VOCs) are a class of carbon-based chemicals known for there ability to evaporate easily at room temperature (Carroll and Kirschman, 2022). VOCs can be produced by human activities, with the primary anthropogenic

sources being vehicle emissions, industrial processes, building materials, solvents, personal car products, the petroleum industry, and vehicular transport (Hester and Harrison (1995) - McDonald et al. (2018) - Rajabi et al. (2020)). VOCs are considered as one of the most important precursors in the formation of tropospheric ozone and secondary organic aerosols (Atkinson and Arey, 2003). These chemicals play a crucial role in ground-level photochemical ozone formation by controlling oxidant production rate in areas with sufficient $NO_x$ (Nitrogen Oxides) concentrations (Hester and Harrison, 1995). On a global scale,

VOCs are mainly emitted from natural sources: soils, oceans and vegetation. The VOC flux emitted from terrestrial vegetation accounts for 90% of the total emission (Guenther et al., 1995). Quantitatively, the most important biogenic volatile organic compound (BVOC) is isoprene ($C_5H_8$). According to MEGANv2.1 (Model of Emission of Gases and Aeorosols from Nature version 2.1) (Guenther et al., 2012), isoprene accounts for about half of all the biogenic species emitted. Isoprene is also known for its high reactivity, as it contributes considerably to the formation of ground-level ozone (Chameides et al., 1988).

Monoterpenes and sesquiterpenes are also considered as important BVOCs due to their substantial impact on the generation of atmospheric organic aerosols on a global scale (Griffin et al. (1999) - Ervens et al. (2011) - Shrivastava et al. (2017)). The emission of ozone and the formation of atmospheric aerosols have effects that reach beyond air quality and human health concerns. They also exert a substantial influence on the current and future state of our climate. Consequently, achieving a precise estimation of BVOCs is of utmost importance. This precision is also crucial for making accurate forecasts of air pollutants

using chemical-transport models on both regional and global scales. Such precise predictions are not only fundamental for assessing air quality but also for quantifying the exact radiative forcing effects arising from ozone and aerosols under both present and future climate conditions. In this context, biogenic emissions are expected to alter in the future as a response to the changing patterns of temperature, solar radiation, land cover and use, and the increasing frequency and intensity of drought events. This creates a need for BVOC modelling tools that can be applied to study present and future climate and air quality

modelling assessment.

The terrestrial BVOC model used in the present study is MEGANv2.1, which is one of the most used models within the biogenic emissions and atmospheric chemistry community to estimate the flux of biogenic organic compounds. It can be used in an offline mode but has also been coupled with another models. Several studies have been conducted implementing the MEGAN model within various canopy environment models or chemical-transport models, each model has a different version/implemen-

tation of the MEGAN algorithms and different weather and land cover driving variables. As a result, the estimated emissions can differ considerably (the annual global isoprene emission varies between 311Tg and 637Tg) (Messina et al. (2016) - Henrot et al. (2017) - Bauwens et al. (2018) - Zhang et al. (2021)).





Our scientific aim was to derive a method for estimating BVOC emissions for present and future climate scenarios that would be capable of considering both atmosphere and land surface processes as well as land-atmosphere interactions that impact vegetation. Our objective was to therefore develop a modelling system for BVOCs based on MEGANv2.1 that would be flexible enough to use a variety of meteorological forcing datasets, e.g., present day reanalyses and output from climate models for future scenarios. Furthermore, this modelling system would have to be capable of simulating impacts on vegetation arising from atmosphere-land interactions. In this study, we have therefore chosen to implement MEGANv2.1 within the SURFEXv8.1 (Surface Externalised) model, which is a land surface modelling platform developed by Météo-France in cooperation with the scientific community. While MEGANv2.1 has been coupled with SURFEX in previous work, this was done in the frame of the mesoscale atmospheric model MESO-NH (Lac et al., 2018) that includes online coupled chemistry. We have been motivated to develop this coupling further for the following reasons. First, SURFEX can be used in offline mode (i.e. using an external meteorological forcing file), this option enables simulations to be performed in present and future climates. Second, SURFEX includes a detailed canopy environment model called ISBA (Le Moigne, 2018). This scheme provides precise vegetation-type-dependent parameters such as soil moisture, Leaf Area Index (LAI), vegetation fraction, temperature, etc. Additionally, this scheme can simulate LAI, which varies in parallel with numerous environmental and meteorological variables. Based on this dynamic LAI, the coupled model can assess and predict the impact of climate change on the biosphere. This impact primarily include alterations in the density and distribution of vegetation, thereby exerting a direct influence on the release of biogenic compounds. In this respect, coupling the SURFEX and MEGAN models can create a feedback loop that takes into account both the impact of climate on vegetation and the impact of vegetation on climate.

The SURFEX and MEGAN2.1 models are presented in section 2, as well as a description of the models offline coupling. Section 3 is dedicated to the evaluation of the coupled model isoprene emissions in comparison with other isoprene inventories. The evaluation of the sensitivity tests results conducted on MEGAN's driving variables is discussed in section 4.

## 2   Models description

### 2.1   SURFEX model

SURFEX (Surface Externalisée, in French) (Le Moigne, 2018) is a surface modelling platform developed by Météo-France in cooperation with the scientific community. SURFEX simulates the interaction between the surface and the atmosphere by simulating the flux exchange between the soil and the upper atmospheric layer (e.g., latent heat flux, sensible heat flux, $CO_2$ flux, chemical species and aerosols). The most recent version of SURFEX (SURFEXv9.0) was released in January 2023; however, in this work, we have used SURFEXv8.1 which is widely used at present (Schoetter et al. (2020) - Zsebeházi and Szépszó (2020) - Schoetter et al. (2017)).

SURFEX can be run in an offline mode or coupled to an atmospheric model, e.g., the global numerical weather prediction model ARPEGE (Déqué et al., 1994). Used in an online mode, SURFEX extracts the necessary meteorological data from the global weather prediction model. In offline mode, a forcing file should be prescribed as input to the model. The forcing file should contain spatio-temporal gridded maps of atmospheric variables: air temperature, specific humidity, wind components,





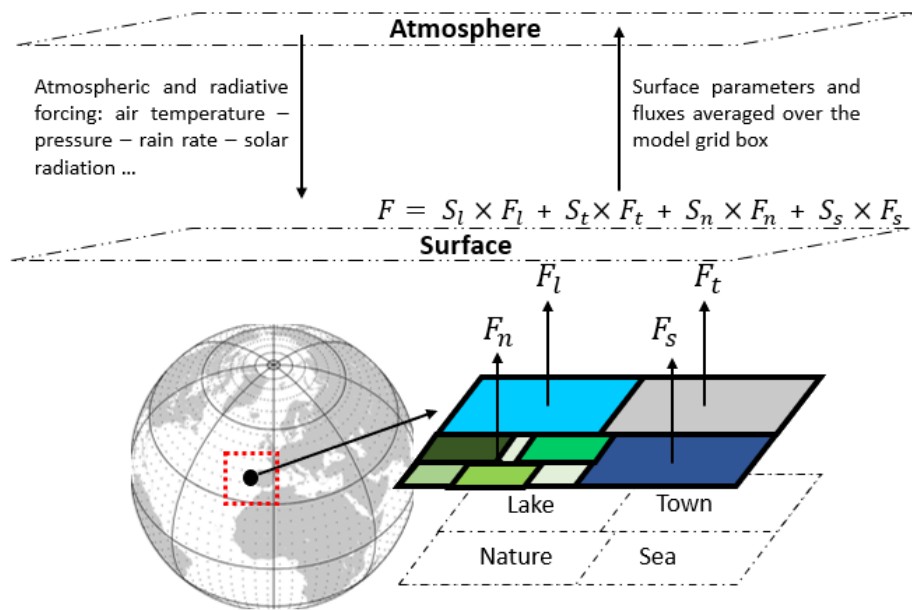

**Figure 1.** Grid cell representation in SURFEX and description of flux exchanges between the surface and atmospheric layer above.

pressure, rain rate, $CO_2$ and radiative variables: solar radiation and infrared radiation. During a model time step, each surface grid box receives the forcing variables listed above, in return SURFEX computes averaged fluxes for momentum, sensible and latent heat, chemical species and dust fluxes, etc, and then returns these quantities to the atmosphere by adding radiative terms such as surface temperature, direct and diffuse surface albedo, and surface emissivity (Le Moigne, 2018).

As shown in Figure 1, each grid box in SURFEX is represented by 4 adjacent tiles: nature, urban areas, sea or ocean and lakes. The final fluxes are the average of the fluxes calculated over nature, city, sea/ocean and lake, weighted by their respective fraction ($S_l$, $S_t$, $S_n$, $S_s$). SURFEX contains four principal surface schemes: ISBA for the nature tile (Calvet et al., 1998), TEB for urban areas (Masson, 2000), FLAKE Mironov et al. (2010) for lakes and SEA for sea and oceans. SURFEX can also simulate aerosol chemistry and surface processes, and can be used for assimilation of surface variables (Le Moigne, 2018).

To define the surface coverage, SURFEX uses ECOCLIMAP-II, which is a 1km global database of land covers made by CNRM (Centre National des Recherches Météorlogiques, in French) (Faroux et al., 2013). It describes the types of surfaces covering the whole earth.

ECOCLIMAP-II provides the fraction data for the 19 patches (nature tile). In addition to that, it provides land surface parameters relative to each patch, i.e., each vegetation type has a defined soil depth, height of trees, LAI (Leaf Area Index) available at

10 day time steps and vegetation fraction. LAI is represented by a 5-year averaged LAI climatology over the period 2002-2006. In ISBA, the calculation of surface parameters is based on an aggregation process at patch level (i.e., from the 19 land cover types down to the selected number of patches) for each point of the grid according to the horizontal resolution (Le Moigne, 2018).



## 2.2 MEGANv2.1 model

The MEGAN model is a global emission platform designed to estimate the net emission of gases and aerosols from terrestrial ecosystems into the atmosphere. It is an updated version of MEGANv2.0 developed by Guenther et al. (2006) to estimate isoprene flux and MEGAN2.02 which was described for monoterpene and sesquiterpene emissions by Sakulyanontvittaya et al. (2008).

MEGANv2.1 (the model's routines and input data can be found here https://bai.ess.uci.edu/megan/data-and-code/megan21,
last access: 8 September 2023) includes algorithms that take into account the main known processes controlling biogenic emissions, it allows to estimate the flux of 19 compound classes, which are decomposed into 147 individual species such as isoprene, monoterpenes, sesquiterpenes, carbon monoxide, alkanes, alkenes, aldehydes, acids, ketones, and other oxygenated VOCs (Guenther et al., 2012). Those species can then be lumped into the appropriate categories for the chemical scheme for use in chemical transport models. The stand-alone version of MEGANv2.1 requires as input weather data (temperature,
precipitation, solar radiation, wind, photosynthetic photon flux), atmospheric chemical composition ($CO_2$ concentration), land cover data (plant functional types distribution and LAI data) and emission factor data.

The estimation of biogenic fluxes in MEGANv2.1 is based on a simple equation (Equation 1) to calculate the net primary emission flux from terrestrial landscapes ( $F_i$ ) into the above-canopy atmosphere ($\mu m^{-2} s^{-1}$). This equation comprises two significant components: firstly, the emission factor, which represents the emission potential of a specific compound associated
with a particular vegetation type, and secondly, the emission activity factor, which reflects how this emission potential responds to variations in environmental conditions and meteorological conditions.

$$F_i = \gamma_i \times \sum_{j=1}^{n} (\varepsilon_{ij} \times \chi_j) \tag{1}$$

Where $\gamma_i$ is the dimensionless activity factor of a compound $i$ (this factor is equal to 1 in standard conditions described below), $\varepsilon_{ij}$ is the emission potential (also known as emission factor) of a compound $i$ and vegetation type $j$ at standard conditions and
$\chi_j$ is the fractional grid box areal coverage.

### 2.2.1 Vegetation and emission factor

A grid cell in MEGANv2.1 is represented by different types of vegetation also called Plant Functional Types (PFTs). A distribution of 16 PFTs is used to represent the vegetation cover, consistent with the vegetation categories used in the Community Land Model version 4 (CLM4) (Gent et al., 2011), which is a model used to simulate the interactions between the surface and
the atmosphere.

The emission factor represents the potential of a vegetation type to emit a specific chemical species under standard conditions. The list of standard conditions used in MEGANv2.1 is shown in Table 1. These conditions are relative to vegetation (e.g., LAI, growing and mature foliage fractions), meteorology (e.g., solar angle, PPFD transmission, temperature, humidity, wind speed), soil (e.g., soil moisture) and canopy (e.g., the past 24h and 240h temperature and PPDD for sun and shade leaves).
The estimation of BVOCs in MEGANv2.1, can be done by using global gridded high-resolution emission potential map pre-





| Parameter | Standard value |
|---|---|
| LAI | $5\ m^2 m^{-2}$ |
| Canopy | 80% mature, 10% growing and 10% old foliage. |
| Solar angle | 60° |
| PPFD transmission | 0.6 |
| Air temperature | $303\ K$ |
| Humidity | $14\ kg\ g^{-1}$ |
| Wind speed | $3\ m\ s^{-1}$ |
| Soil moisture | $0.3\ m^3\ m^{-3}$ |
| Temperature of the past 24 and 240h | $297\ K$ |
| PPFD of the past 24h and 240h | $200\ \mu mol\ m^{-2}\ s^{-1}$ for sun leaves and $50\ \mu mol\ m^{-2}\ s^{-1}$ for shade leaves. |

**Table 1.** List of standard conditions used in MEGANv2.1 (Guenther et al., 2006).

scribed as input to the model (this map is provided with the MEGAN code for 10 predominant biogenic species) or by using
PFTs spatial coverage and PFTs dependent emission potential data.

### 2.2.2 Emission activity factor

The emission activity factor represents the response of the vegetation to a change in environmental and meteorological condi-
tions. The activity factor $\gamma_i$ of a compound class **i** is calculated in the MEGANv2.1 fortran code as the multiplication of factors
accounting for emission response to light $\gamma_{P,i}$ , temperature $\gamma_{T,i}$  leaf age $\gamma_{A,i}$  soil moisture $\gamma_{SM,i}$  leaf area index (LAI) and
$CO_2$ inhibition $\gamma_{CO2,i}$ as follows:

$$\gamma_i = C_{CE} \times LAI \times \gamma_{P,i} \times \gamma_{T,i} \times \gamma_{A,i} \times \gamma_{SM,i} \times \gamma_{CO2,i} \tag{2}$$

The canopy environment coefficient $C_{CE}$ is used to normalise the activity factor at standard conditions listed above and is
dependent on the canopy environment model being used. In MEGANv2.1 code, the equation used to calculate $\gamma_i$ is:

$$\gamma_i = \gamma_{A,i} \times \gamma_{SM,i} \times \gamma_{CO2} \times ((1 - LDF) \times \gamma_{TLI,i} \times \gamma_{LAI,i} + LDF \times \gamma_{TLD,i}) \tag{3}$$

Where $\gamma_{TLI,i}$ is the sum of temperature light-independent activity factor at 5 canopy levels and $\gamma_{TLD,i}$ is the sum of the product
of light activity factor and temperature light-dependent activity factor at 5 canopy levels. In fact, in MEGANv2.1 the emission
of each compound class includes a light-dependent fraction (LDF) and a light-independent fraction (LIF = 1 – LDF) that is
not influenced by light. Each compound has a specific LDF (for isoprene LDF = 1). Light-dependent emissions are calculated
following the isoprene response to temperature described by Guenther et al. (2006) and light-independent emissions follows
the monoterpene exponential temperature response described by Guenther et al. (1993). The calculation of light-dependent
and independent factors is based on a detailed canopy environment model that estimates light (PPFD), temperature (T), and
fraction of sun and shade leaves at 5 canopy levels. The calculation of $\gamma_{TLI,i}$ and $\gamma_{TLD,i}$ is presented in equations 4, 5, 6



and 7, where $\gamma^j_{TLI}$ and $\gamma^j_{TLD}$ are calculated as the sum of temperature light-independent factor and light-dependent factor respectively weighted by the fraction of sun leaves $f^j_{sun}$ and the fraction of shade leaves $(1 - f^j_{sun})$ in each canopy level.

$$\gamma_{TLI} = \sum_{j=1}^{5} \gamma^j_{TLI} \tag{4}$$

$$\gamma^j_{TLI} = \gamma_{TI,sun} \times f^j_{sun} + \gamma_{TI,shade} \times (1 - f^j_{sun}) \tag{5}$$

$$\gamma_{TLD} = C_{CE} \times LAI \times \sum_{j=1}^{5} \gamma^j_{TLD} \tag{6}$$

$$\gamma^j_{TLD} = \gamma_{P,sun} \times \gamma_{TD,sun} \times f^j_{sun} + \gamma_{P,shade} \times \gamma_{TD,shade} \times (1 - f^j_{sun}) \tag{7}$$

The calculation of $\gamma_{TI,sun}$, $\gamma_{TI,shade}$, $f^j_{sun}$, $\gamma_{P,sun}$, $\gamma_{TD,sun}$, $\gamma_{P,shade}$ and $\gamma_{TD,shade}$ is detailed in Guenther et al. (2012).

## 2.3 SURFEX-MEGAN coupling

The coupling of MEGAN2.1 and SURFEXv8.1 is based on a previous implementation of MEGAN in MESO-NH5.4. MESO-NH5.4 is an atmospheric non-hydrostatic research model designed for studies of physics and chemistry (Lac et al., 2018). This coupling involved merging MEGAN routines and linking the required inputs of the biogenic model with SURFEX's parameters.

The present study focuses on the online integration of MEGAN in SURFEX. The ultimate aim of this coupling is to be able to force the coupled model through various climate change scenarios in order to assess climate change impact on the biosphere and to quantify the effect of these changes on biogenic emissions and therefore on global and local air quality. As well, this coupling aims to improve biogenic emission estimations by providing the MEGAN model with detailed vegetation-dependent inputs at patch level. This allows key land surface parameters used by MEGAN, i.e. leaf area index and soil moisture, to be calculated at a more precise scale. Thus, activity factors are individually calculated for each patch. This approach allows for a more accurate representation of biogenic emissions in the context of climate change and their impact on air quality.

In the coupled model the estimation of biogenic fluxes of various species was carried out based on 16 vegetation types extracted from the ECOCLIMAP-II database (Faroux et al., 2013). Each vegetation type from ECOCLIMAP-II was mapped to its corresponding defined in CLM4. Table 2 represents the mapping used in the coupled model. For most CLM4 PFTs, existing similar vegetation types are defined in ECOCLIMAP-II. However, when considering shrubs, CLM4 classifies them into three distinct categories: Evergreen temperate shrub, Deciduous temperate shrub, and Broadleaf deciduous shrub. Conversely, ECOCLIMAP-II does not provide separate classifications for these three distinct types of shrubs. To overcome this limitation,





| CLM PFT number | Description | ECOCLIMAP patch number | Description | Type |
|---|---|---|---|---|
| 1 | Needleleaf Evergreen Temperate Tree | 15 | Temperate Needleleaf Evergreen | NT |
| 2 | Needleleaf Evergreen Boreal Tree | 5 | Boreal Needleleaf Evergreen | NT |
| 3 | Needleleaf Deciduous Boreal Tree | 17 | Boreal Needleleaf Cold-Deciduous Summergreen | NT |
| 4 | Broadleaf Evergreen Tropical Tree | 6 | Tropical Broadleaf Evergreen | BT |
| 5 | Broadleaf Evergreen Temperate Tree | 14 | Temperate Broadleaf Evergreen | BT |
| 6 | Broadleaf Deciduous Tropical Tree | 13 | Tropical Broadleaf Deciduous | BT |
| 7 | Broadleaf Deciduous Temperate Tree | 4 | Temperate Broadleaf Cold-Deciduous Summergreen | BT |
| 8 | Needleleaf Broadleaf Deciduous Boreal Tree | 16 | Boreal Broadleaf Cold-Deciduous Summergreen | NT |
| 9 | Broadleaf Evergreen Temperate Shrub | 19 | Shrub [-30° < lat < 30°] | SHRB |
| 10 | Broadleaf Deciduous Temperate Shrub | 19 | Shrub [-60° < lat < -30° or 30° < lat < 60°] | SHRB |
| 11 | Broadleaf Deciduous Boreal Shrub | 19 | Shrub [60° < lat] | SHRB |
| 12 | Arctic C3 Grass | 18 | Boreal Grass | GRLD |
| 13 | Cool C3 Grass | 10 | Grassland (C3) | GRLD |
| 14 | Warm C4 Grass | 11 | Tropical Grassland (C4) | GRLD |
| 15 | Crop1 (Wheat) | 7 | C3 Cultures Types | CROP |
| 16 | Crop2 (Corn) | 8 | C4 Cultures Types | CROP |

**Table 2.** Description of the mapping between CLM4 and ECOCLIMAP vegetation types, the 16 PFTs are grouped into 6 vegetation types (NT: Needleleaf Trees, BT: Broadleaf Trees, SHRB: Shrubs, GRLD: Grassland, CROP: Crops) (5 ECOCLIMAP patches are not included in this list as they represent patch 1 = bare soil, patch 2 = rock, patch 3 = snow, patch 9 = irrigated crops and patch 12 = peat bogs, parks and gardens).

the three plant functional types corresponding to the different types of shrubs were introduced within ECOCLIMAP-II by assigning the shrub patch to a specific geographical area based on a given latitudinal range. The evergreen temperate shrub type is specified in the coupled model as the shrub patch in tropical regions (-30° < latitude < 30°), the deciduous temperate shrub in temperate regions (-60° < latitude < -30° or 30° < latitude < 60°), and the deciduous boreal shrub in boreal regions (60° < latitude). This approach allows for a more accurate representation of shrubs in the coupled model.

Figure 2 represents a comparison between the vegetation types used in MEGAN stand-alone and the ones defined in ECOCLIMAP-II. For comparison, we have grouped the 16 PFTs into 6 main vegetation types: Broadleaf evergreen trees, Needleleaf evergreen trees, Deciduous trees, Shrubs, Grassland and Crops. The vegetation spatial distribution and intensity is similar for most vegetation types in ECOCLIMAP-II and CLM4. For shrubs, the substantial difference in vegetation distribution is due to the vegetation height threshold used in ECOCLIMAP-II (2m) and in CLM4 (10m). For vegetation-related input data, MEGAN can use climatological LAI from the ECOCLIMAP-II database, in this case, the LAI is defined for each vegetation type in a 10-day time step or the dynamic LAI estimated for each vegetation type with the vegetation scheme in SURFEX. $LAI_v$ defined as the LAI in a grid cell divided by the vegetation fraction is considered equal to the current LAI and $LAI_p$ (previous LAI) is



**Figure 2.** Spatial coverage of the 6 vegetation types defined in Table 2: BT (Broadleaf Trees), NT (Neadleleaf Trees), DT (Deciduous Trees), GRLD (Grassland), SHRB (Shrubs) and CROP (Crops) in CLM4 (right) and in ECOCLIMAP-II (left).



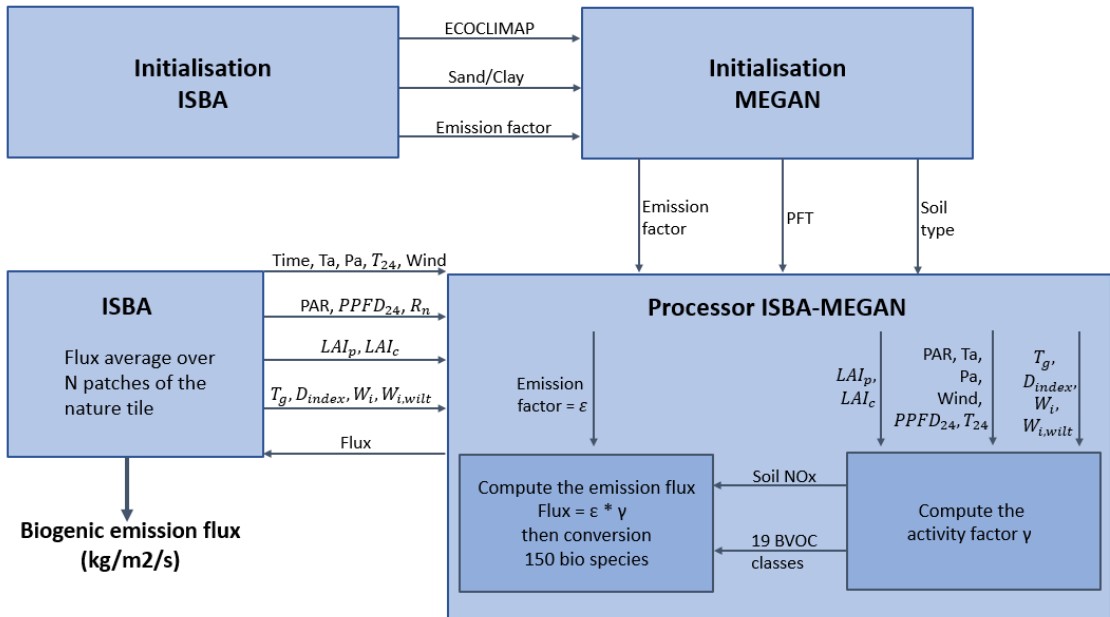

**Figure 3.** Schematic description of the SURFEX-MEGAN coupling. $T_a$ is the temperature at 2m height, $P_a$ is the surface pressure, $T_{24}$ and $PPFD_{24}$ are the previous day mean temperature and PPFD, respectively, $W_i$ and $W_{i,wilt}$ are the soil moisture and wilting point at different soil layers, respectively, $T_g$ is the soil surface temperature, $\epsilon$ is the emission factor, $\gamma$ is the activity factor, $R_n$ is the incoming shortwave solar radiation flux, $LAI_p$ and $LAI_c$ are the LAI value of the previous and current day, respectively, $D_{index}$ is the soil category.

200    defined as the LAI value of the past 10 days.

In the SURFEX model time step, all surface variables are interpolated and updated for each grid cell. Each tile is treated independently by using a specific scheme. For the Nature tile, the surface parameters are calculated following the vegetation-type aggregation process, which merges several vegetation types into a single patch (ranging from 1 to 19).

It is important to clarify that the coupling of SURFEX and MEGAN is online, which means that MEGAN's estimation of
205    biogenic fluxes interact dynamically with the ISBA scheme (Interaction between Soil Biosphere and Atmosphere). ISBA is the scheme used for Nature tile to compute the exchanges of energy and water between the continuum soil-vegetation-snow and the atmosphere above.

The online implementation of MEGAN was done following the SURFEX's conceptual framework, which separates the initialisation phase from the temporal evolution phase. This involved setting up specific routines to initialise and interpolate
210    MEGAN-related parameters (e.g., emission factors). The temporal estimation of biogenic emissions was carried out as an integral part of the ISBA scheme. This was achieved by integrating MEGAN routines that estimate the activity factor for each vegetation patch, using vegetation parameters estimated by ISBA, which encompass factors like soil moisture and wilting point at different layers (depending on the soil discretization method), leaf area index, photosynthetic active radiation (PAR), surface temperature, etc. Figure 3 shows a global representation of the online implementation of MEGAN in SURFEX.





## 3 Evaluation of SURFEX-MEGAN flux estimates

### 3.1 Model setup

The coupled SURFEX-MEGAN model was utilised to conduct a global simulation of isoprene in 2019 using ERA5 meteorological forcing. This simulation is referred to as the reference simulation (abbreviated to REF).

ERA5 is a reanalysis based on the integrated forecasting system IFS (numerical weather forecasting model and data assimilation system, developed jointly by ECMWF and by Météo-France) (Hersbach et al., 2020). For the REF SURFEX-MEGAN simulation, the ERA5 forcing file includes hourly reanalysis meteorological fields defined on a $1° \times 1°$ spatial resolution grid (re-gridded from the native 31 km $\times$ 31 km resolution). Temperature and specific humidity were extracted at 2m height; wind speed and wind direction were calculated based on zonal and meridian wind components at 10m height. As there are no available inputs for surface incident diffuse shortwave radiation and $CO_2$ rate, these parameters were assigned values of $0 \; \mathrm{Wm}^{-2}$ and $410 \; \mathrm{ppm}$, respectively. The $CO_2$ concentration value corresponds to the 2019 annual mean of $CO_2$ observed at Mauna Loa (Keeling et al., 2000).

In this study, the calculation of PPFD and temperature for sun and shade leaves at different canopy heights was done using the canopy model integrated in MEGAN; the incoming PAR (Photosynthetically active radiation) at the top of the canopy was assumed to be 48% of the incoming shortwave radiation (Jacovides et al., 2003) (Nagaraja Rao, 1984); a conversion factor of 4.6 and 4.0 $\mu_{mol} \; photons \; J^{-1}$ was used to convert PAR to PPFD for diffused and direct radiation respectively (Guenther et al., 2012). Unless otherwise stated, in all coupled model simulations the estimation of isoprene flux was done based on isoprene potential map and the effect of soil moisture deficit and $CO_2$ on BVOC emissions was not taken into account (the $\gamma_{sm}$ and $\gamma_{CO2}$ factors were assigned to 1). This choice allows a better comparison with other emission inventories.

For simplicity, we have used the ISBA 2-L scheme in the present study. In this scheme, the soil is represented with two layers, the heat and moisture exchanges between the layers and the atmosphere is modelled with the force-restore method (Le Moigne, 2018), this approach is described further in section 4.

### 3.2 Comparison of SURFEX-MEGAN isoprene emissions with other datasets

#### 3.2.1 Isoprene inventories description

The validation of the results obtained by the coupled model was evaluated by comparing the 2019 global and regional isoprene emission results with other isoprene inventories estimated with the MEGAN model. The data used for this comparison are presented in Table 3, additional information regarding the simulation setup used to generate the results is also provided. For inventories with unavailable 2019 isoprene emissions, the closest available year was used for comparison.

CAMS-GLOB-BIO is a high-resolution global emission inventory of the main biogenic species including isoprene, monoterpene, sesquiterpenes, methanol, acetone, and ethene (Sindelarova et al., 2022). It provides monthly average inventories and monthly average daily profiles of 3 different emission scenarios for the period 2000-2019. CAMS-GLOB-BIOv1.2 is a $0.5° \times 0.5°$ spatial resolution dataset obtained with ERA-interim meteorology, the vegetation cover is based on the CLM4 16 PFTs





and the emissions are calculated based on the emission potential map provided along with the MEGANv2.1 code. CAMS-GLOB-BIOv3.0 and CAMS-GLOB-BIOv3.1 have a higher spatial resolution of $0.25° \times 0.25°$ and are based on the ERA5 meteorology. The aim of the 3.0 scenario is to capture the impact of the land cover annual evolution on biogenic emissions

by using the land cover data provided by the Climate Change Initiative of the European Space Agency (ESA-CCI). The 3.1 scenario uses the CLM4 vegetation cover and emission potential map for isoprene and main monoterpenes. The EP (Emission Potential) map was updated over Europe using high-resolution land cover maps and detailed information of tree species composition and emission factors from the EMEP MSC-W model system.

MEGAN-MACC is a biogenic emission inventory developed under the Monitoring Atmospheric Composition and Climate

project (MACC) (Sindelarova et al., 2014). It includes monthly mean emissions of 22 biogenic species (isoprene, monoterpenes, sesquiterpenes, methanol and other oxygenated VOCs and carbon monoxide) estimated by the MEGANv2.1 model on a global $0.5° \times 0.5°$ grid for the time period 1980-2020, using meteorological fields of Modern-Era Retrospective Analysis for Research and Applications (MERRA).

The ALBERI dataset is a bottom-up inventory of isoprene emissions developed in the frame of the ALBERI project funded

by the Belgian Science Policy Office (Opacka et al., 2021). Isoprene emissions are estimated by the MEGANv2.1 model, coupled with the canopy environment model MOHYCAN (Model for Hydrocarbon emissions by the CANopy) (Wallens, 2004) (Bauwens et al., 2018). The model was driven by the ERA-interim meteorological fields, vegetation description was provided from satellite-based Land Use/Land Cover (LULC) datasets at annual timesteps. The LULC datasets are based on the MODIS PFT dataset and are adjusted to match the tree cover distribution from the Global Forest Watch (GFW) database (Hansen et al.,

265   2013).

### 3.2.2   Spatio-temporal distribution analysis

The global annual isoprene emission estimated with SURFEX-MEGAN simulation is 443Tg. The isoprene estimates of the coupled model falls within the range of previous reported values calculated with MEGANv2.1, varying between 311Tg and 637Tg. The discrepancies between isoprene totals obtained by different studies are due to many factors, including model as-

sumptions and input data (e.g., meteorology, LAI, vegetation distribution). In fact, according to Messina et al. (2016), isoprene emissions are highly dependent on LAI , as they linearly increase up to a LAI=$2m^2/m^2$, then gradually decrease to become almost constant above $5m^2/m^2$. As shown by Sindelarova et al. (2014), the use of different LAI inputs (MERRA reanalysis data instead of MODIS LAI data) can lead to a 4% increase in annual isoprene emissions. The use of different data of photosynthetically active radiation (PAR) can also significantly impact the calculated isoprene emissions. Sindelarova et al. (2014) found

that using PAR calculated from incoming shortwave radiation instead of PAR from the MERRA reanalysis led to a 17.5% increase in total isoprene emissions. Further in this section, we will examine other individual factors responsible for the total isoprene discrepancies and the differences in spatio-temporal distribution between isoprene estimates from SURFEX-MEGAN and other isoprene inventories.

Figure 4 displays the mean annual isoprene flux of the six inventories. As shown in Figure 4, the spatial distribution of isoprene

shows similar general spatial patterns for the different datasets, with important isoprene emissions located in South America





| Ref. | Dataset | Resolution | Weather | PAR | LAI | PFT | Emission potential | Data availability | Isoprene Tg/year |
|---|---|---|---|---|---|---|---|---|---|
| a | CAMS-GLOB-BIOv1.2 | $0.5° \times 0.5°$ | ERA Interim | $\alpha$ | MODIS | CLM4 | EP map | 2000-2018 | 385 (2018) |
| a | CAMS-GLOB-BIOv3.0 | $0.25° \times 0.25°$ | ERA5 | $\alpha$ | MODIS | ESA-CCI | PFT dependent | 2000-2019 | 311 (2019) |
| a | CAMS-GLOB-BIOv3.1 | $0.25° \times 0.25°$ | ERA5 | $\alpha$ | MODIS | CLM4 | EP map(updated in Europe) | 2000-2020 | 471 (2019) |
| b | MEGAN-MACC | $0.5° \times 0.5°$ | MERRA | MERRA | MODIS | CLM4 | EP map | 1980-2020 | 637 (2019) |
| c | ALBERI | $0.5° \times 0.5°$ | ERA Interim | ERA Interim | MODIS | CLM4 | PFT dependent | 2001-2018 | 347 (2018) |
| This study | SURFEX-MEGAN | $1° \times 1°$ | ERA5 | $\alpha$ | ECOC-LIMAP | ECOC-LIMAP | EP map | 2019 | 443 (2019) |

**Table 3.** List of isoprene inventories used for the model validation and description of driving input data, $\alpha$ is the conversion factor used to approximate PAR from surface solar downward radiation = 0.45, (a) Sindelarova et al. (2022), (b) Sindelarova et al. (2014), (c) Opacka et al. (2021).

(the Amazon rainforest) and Africa (the Congo rainforest), however some differences can be discerned in Australia as well as in the maximum isoprene emission estimated by each inventory. These discrepancies, can be attributed to the emission potential data used in each simulation and the PFT cover present in the area, as the spatial distribution of isoprene can be highly impacted by both the model assumptions regarding emission capacity and the spatial distribution of the vegetation types considered.

The isoprene flux in the SURFEX-MEGAN simulation shows a comparable spatial pattern to CAMS-GLOB-BIOv3.1. This similarity can be attributed to the fact that both simulations use ERA5 meteorological forcing, the same isoprene emission potential gridded map and similar vegetation distributions (cf. section 2.3). The isoprene emissions in MEGAN-MACC shows also similar spatial pattern, with more significant emissions located in Australia and South America. In contrast, the spatial distribution of isoprene in CAMS-GLOB-BIOv3.0 and ALBERI differs significantly from that of the SURFEX-MEGAN sim-

ulation, as these two simulations were produced using the PFT dependent emission potential table from MEGAN.

In isoprene low emission regions, such as North America, Europe, and North Asia, isoprene emissions from SURFEX-MEGAN are particularly higher when compared to other isoprene inventories. This discrepancy can be attributed to variations in vegetation types and their intensity between CLM4 and ECOCLIMAP in these specific areas. As shown in Figure 2, needleleaf trees and grassland density in Asia and North America are notably greater in ECOCLIMAP, making the emissions in these regions

substantially higher in SURFEX-MEGAN compared to other CLM4 PFTs-based isoprene inventories.



**Figure 4.** Spatial distribution of annual mean isoprene (kg/m²/s) of CAMS-GLOB-BIOv2.1, CAMS-GLOB-BIOv3.0, CAMS-GLOB-BIOv3.1, MEGAN-MACC, ALBERI and SURFEX-MEGAN in 2019 (2018 for CAMS-GLOB-BIOv2.1 and ALBERI).

Figure 5 represents the time series of global monthly isoprene in 2019 of SURFEX-MEGAN compared to the five other inventories. The monthly variation of isoprene emissions in the SURFEX-MEGAN simulation is marked by small monthly fluctuations. The maximum isoprene emission occurs in boreal summer (July/August) with a total isoprene of 40Tg and the minimum in boreal winter (February) with a total isoprene of 33Tg. The annual cycle of SURFEX-MEGAN isoprene is in agreement with the ALBERI and CAMS-GLOB-BIOv1.2 datasets. A visible shift is noticed for MEGAN-MACC and CAMS-GLOB-BIOv(3.0 - 3.1) isoprene annual cycle with peak concentrations occurring in December/January and minimum in May/June.






Figure 6 and 7 represent respectively, monthly and yearly relative contribution of different zonal regions to isoprene emissions for the different datasets. In the SURFEX-MEGAN simulation, the annual cycle of isoprene follows the seasonal cycle: In Boreal summer (May – June – July - August), isoprene emissions are preponderant in the northern hemisphere (60% of total emission in this period) and in austral summer (October - November – December – January – February), isoprene emissions are preponderant in southern hemisphere (64% of total emission in this period). As shown in Figure 6, southern and northern tropical regions predominate throughout the year, their contribution to the total emission in the SURFEX-MEGAN simulation varies between 33% to 60% and 30% to 44% respectively, this is due to the meteorological conditions that are favourable all year-round (both in terms of temperature and solar radiation) and due to the high concentration of vegetation in these areas. Northern temperate regions are only active during boreal summer with a maximum contribution of 24% in July. The contribution of southern temperate regions follows a cyclical pattern, with a maximum in austral summer (6% in reference simulation). Finally, the Arctic is characterised by a very low activity, which is due to the unfavourable weather conditions and relatively low vegetation cover.

The monthly variation in isoprene emissions is strongly influenced by the emitting regions contribution throughout the year. As already mentioned, southern tropical regions are active throughout the year for all isoprene datasets, with particularly high contributions during November/December and lower contributions during June/July. As shown in Figure 7, southern tropical regions account for approximately 49% of annual isoprene emissions in SURFEX-MEGAN and CAMS-GLOB-BIOv2.1. However, their contribution to the annual isoprene is significantly higher in MEGAN-MACC (56%), CAMS-GLOB-BIOv(3.0 - 3.1) (54% - 52%), which can explain the peak in isoprene emissions observed during November/December. Conversely, isoprene emissions from northern temperate regions are relatively higher in SURFEX-MEGAN (10%), CAMS2.1 (9%) and ALBERI (11%), compared to MEGAN-MACC (7%) and CAMS3.0/3.1 (6%). These regions are active mainly during boreal summer, which can explain the isoprene peak observed during July for SURFEX-MEGAN/CAMS2.1/ALBERI.

The isoprene spatial and temporal distribution of the SURFEX-MEGAN coupled model are in agreement with other MEGAN-driven isoprene inventories. The evaluation of the total annual isoprene is however hard to assess, as the emissions are highly affected by both model input data and model assumptions.

## 4 SURFEX-MEGAN isoprene sensitivity tests

In order to analyse isoprene emission variation linked to MEGAN's driving parameters, 3 sensitivity tests were conducted. As stated in (Guenther et al., 2012) isoprene emissions depend on various meteorological and environmental parameters as well as the model assumptions. In this study, we have investigated isoprene emission sensitivity to meteorology using 2 different additional meteorological datasets (both IFS and MERRA) (S1), analysed isoprene emissions with a different set of emission potentials (S2), and studied the impact of soil moisture on isoprene emission (S3). Table 4 summarises the list of sensitivity tests performed in this study, along with a description of each test setup. The impact of each sensitivity test was examined on the global and regional scales by analysing the annual isoprene emission contribution from nine geographical regions defined in the GlobEmission project (www.globemission.eu). The spatial extent of the regions is given in Figure 8.



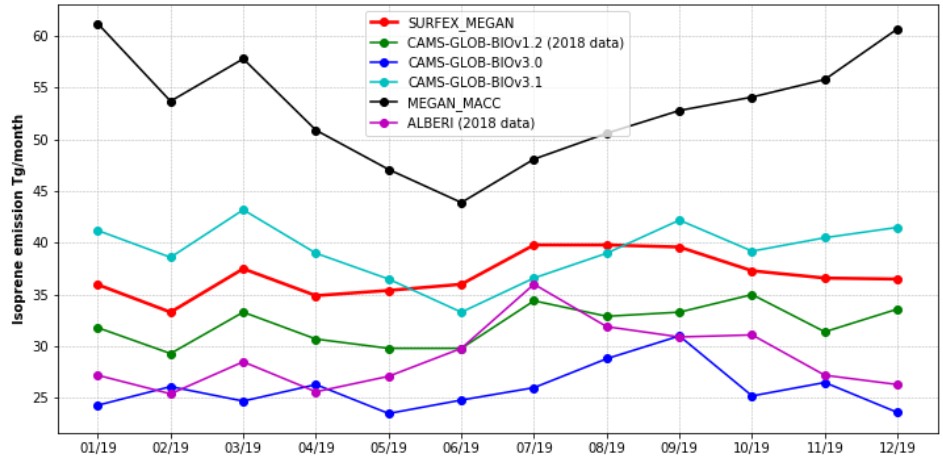

**Figure 5.** Global monthly isoprene (Tg/month) from different the six datasets in 2019. 2018 data was used for CAMS-GLOB-BIOv2.1 and ALBERI.

| Simulation | Description | Meteorology | $\gamma_{SM}$ | Emission potential |
|:---:|:---:|:---:|:---:|:---:|
| RS | reference simulation | ERA5 | =1 | $\epsilon_{map}$ |
| S1 | use of MERRA meteorological forcing | MERRA | =1 | $\epsilon_{map}$ |
| S1 | use of IFS meteorological forcing | IFS | =1 | $\epsilon_{map}$ |
| S2 | use of isoprene emission potential | ERA5 | =1 | $\epsilon_{PFT}$ |
| S3 | study the impact of soil moisture on isoprene | ERA5 | variable | $\epsilon_{map}$ |

**Table 4.** List of sensitivity runs performed.

## 4.1 Meteorology

The emission rate of isoprene can be influenced by a variety of meteorological factors, including temperature, solar radiation and atmospheric humidity. To illustrate the impact of these factors on isoprene emission estimated by SURFEX-MEGAN, two simulations were conducted using two different meteorological datasets: IFS forecast dataset (operational real-time weather forecast, forecast grid data) and MERRA. MERRA was undertaken by NASA's Global Modelling and Assimilation Office. The data were generated with version 5.2.0 of the Goddard Earth Observing System (GEOS) atmospheric model and data assimilation system (DAS) and covers the period from 1979 to present (Rienecker et al., 2011). The MERRA data are defined on an hourly basis on a grid of 0.625° latitude and 0.5° longitude resolution. However to avoid considering the effect of spatial resolution on isoprene emission (Pugh et al., 2013), the MERRA reanalysis meteorological fields were interpolated to align with the reference simulation spatial resolution ( $1° \times 1°$ ).



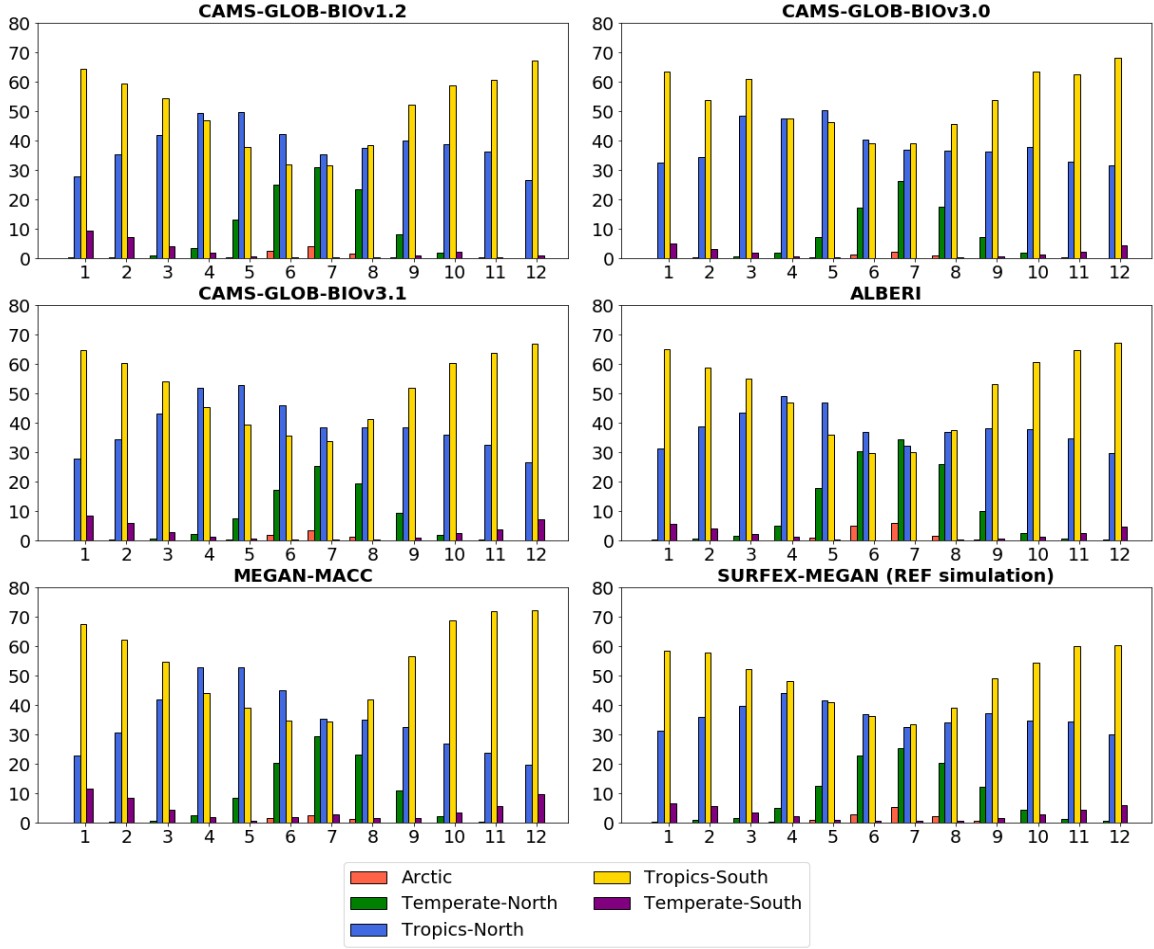

**Figure 6.** Contribution of zonal regions to monthly isoprene in CAMS-GLOB-BIOv2.1, CAMS-GLOB-BIOv3.0, CAMS-GLOB-BIOv3.1, MEGAN-MACC, ALBERI and SURFEX-MEGAN simulation in 2019 (2018 for CAMS-GLOB-BIOv2.1 and ALBERI). The zonal bands are defined as: Artcic (90°,60°), Temperate-North(60°,30°), Tropics-North(30°,0°), Tropics-south(0°,-30°), Temperate-south(-30°,-60°).

The reference simulation uses ERA5 meteorological forcing, however, the version of IFS used in ERA5 is a newer and more advanced version of the IFS that was used in the near real time forecasts in 2019 for operations. This improved version of the IFS for ERA5 uses a numerical climatology model for modelling physical processes, while the version used for operational real-time forecasts uses process parameterization schemes that are optimised for fast and real-time execution. The IFS meteorological forcing was extracted from the IFS operational real-time forecasts model with a spatial resolution of $1° \times 1°$ and a temporal resolution of 3h.





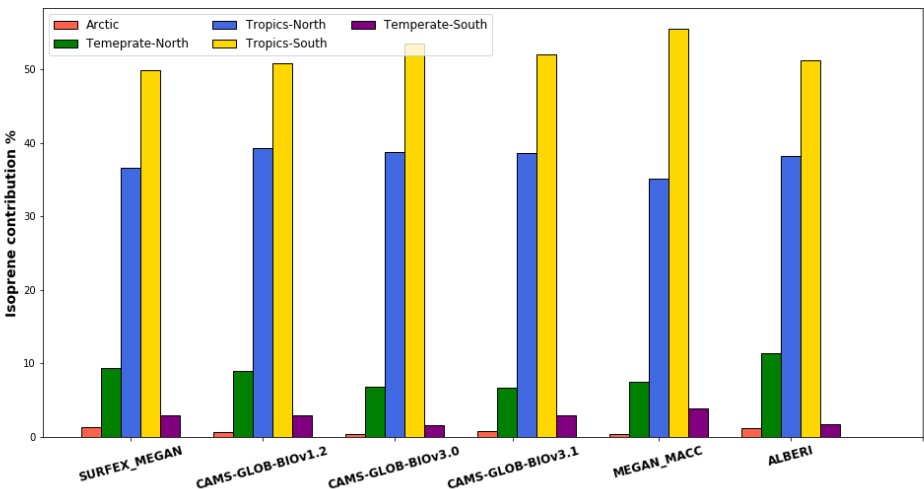

**Figure 7.** Contribution of zonal regions to annual isoprene for different emission datasets in 2019 (2018 for CAMS-GLOB-BIOv2.1 and ALBERI).

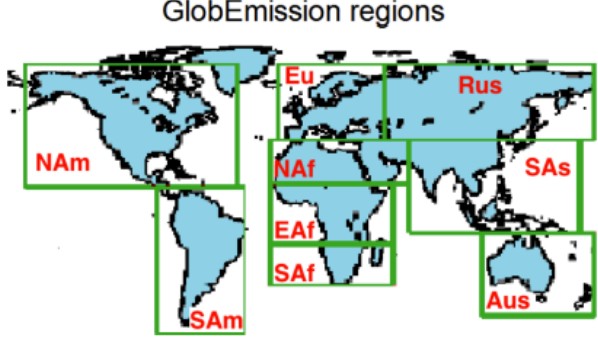

**Figure 8.** Geographical extent of the GlobEmission regions (NAm: North America, SAm: South America, Eu: Europe, NAf: North Africa and middle East, EAf: East Africa, SAf: South Africa, Rus: Russia, SAs: South East Asia, Aus: Australia), from Sindelarova et al. (2014).

The S1-MERRA simulation has the highest global annual isoprene in 2019 with a total of 462Tg, followed by reference simulation (ERA5) and S1-IFS with a total of 443Tg and 421Tg respectively. The mean annual isoprene flux difference in 2019 between S1 simulations and reference simulation is shown in Figure 9. ERA5 isoprene emissions are higher in both the Amazon and Congo rainforests as well as over Indonesia compared to IFS isoprene estimates. On the other hand, ERA5-based isoprene emissions are lower than MERRA isoprene in eastern Australia but higher in Africa. To investigate the origin of these differences, an analysis of meteorological parameters that drive isoprene emission was performed focusing particularly

355





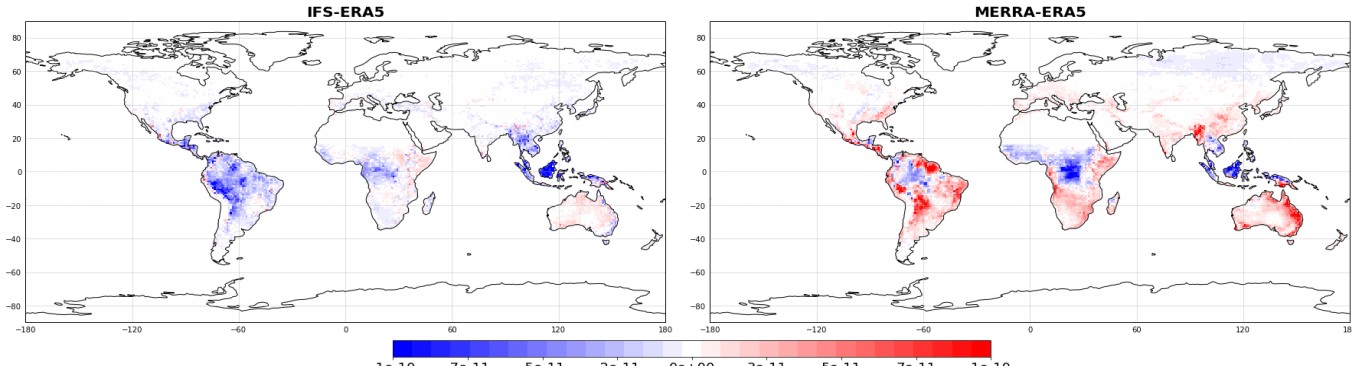

**Figure 9.** Annual mean isoprene difference (kg/m2/s) between S1-IFS and reference simulation (left) and between S1-MERRA and reference simulation (right).

on temperature and solar radiation. These parameters influence the emission of biogenic species via two factors $\gamma_P$ and $\gamma_T$ detailed by Guenther et al. (2012).

As shown by Guenther et al. (2006), the estimate of isoprene flux in MEGAN is temperature dependent, with emissions increasing exponentially with temperature to a maximum that depends on the average temperature of the last 24 hours. MEGAN emissions depend also on the amount of light received by vegetation. The isoprene estimate increases almost linearly with PPFD, the rate of increase depends on the average PPFD over the last 24 hours. To study the linear dependence between the isoprene flux estimates and PPFD, we examined the correlation between the difference in isoprene estimates, and the difference in light (PAR) between the reference (with the ERA5 meteorological forcing) and S1 simulations (with the IFS and MERRA meteorological forcings). Figure 10 displays the temporal correlation coefficient between isoprene flux differences and light differences for reference and S1-IFS simulations, as well as reference and S1-MERRA simulations. The PAR contributes strongly to the explanation of isoprene discrepancies between reference and S1 simulations, as the correlation coefficient exceeds 0.8 in regions where isoprene is emitted. Thus, the difference in isoprene emission flux across the three simulations is mainly due the different PAR input used in the simulation's meteorological forcing file. The correlation study was not conducted on other isoprene meteorological drivers, such as temperature, as the dependence of isoprene on this parameter is exponential.

Figure 11 and Figure 12 represent the isoprene distribution by region of all performed tests and the mean temperature/PAR relative difference between the MERRA and the ERA5 data inputs. On a regional scale, MERRA temperature and downward radiation received by vegetation are higher in Australia and South America compared to ERA5, resulting in higher isoprene estimates in those regions (+10% in Australia and +7% in South America). Conversely, MERRA temperature and radiation inputs are lower in East Africa, resulting in lower isoprene estimates for that region (-3%).

Several studies have been conducted to quantify the MEGAN model sensitivity to meteorology. For example, Arneth et al. (2011) showed that using different meteorological forcings can lead to different emission estimates where the use of CRU



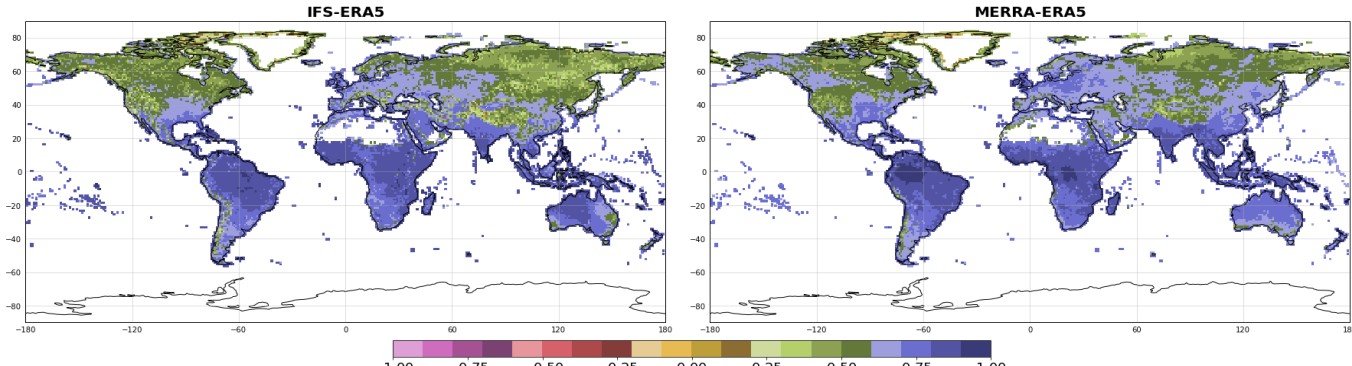

**Figure 10.** Pearson correlation coefficient of PAR and isoprene difference between reference and S1 simulations (S1-IFS (left) and S1-MERRA (right)).

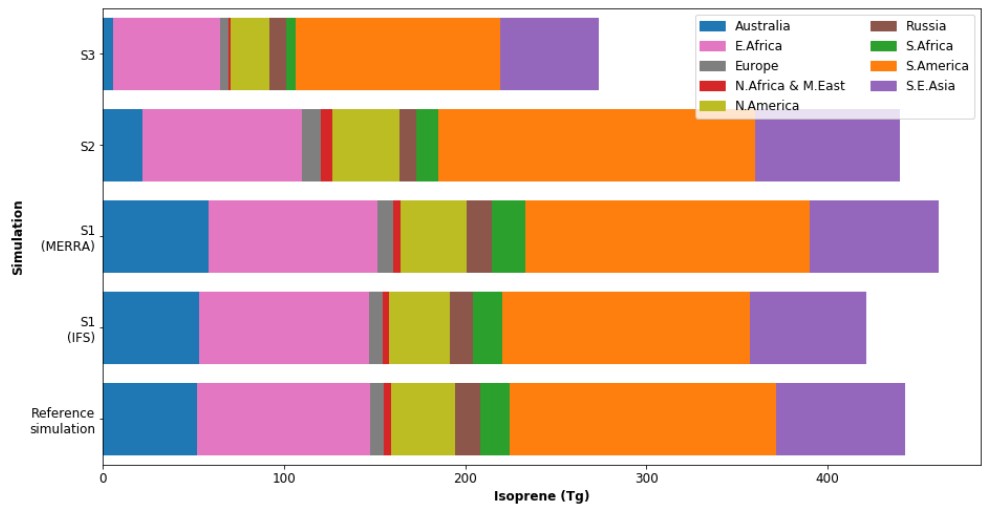

**Figure 11.** Isoprene total emission by region (defined in Figure 8) of the reference simulation, S1 simulation (IFS/MERRA), S2 simulation (emission potential) and S3 simulation (soil moisture).

(Climatic research Unit) meteorology instead of the NCEP (National Center for Environmental Prediction) reanalysis product led to a 10% decrease with MEGANv2. Sindelarova et al. (2022) also detected a difference of the total BVOC MEGANv2.1 estimations between CAMS-GLOB-BIOv2.1 and CAMS-GLOB-BIOv3.1 and explained that the discrepancies are mainly due the use of different meteorological inputs.

On a global scale the use of different meteorological forcing has been observed to have an impact on the amount of isoprene emissions estimated with the SURFEX-MEGAN model. The use of MERRA meteorology led to a 5% increase in isoprene emissions, while the use of IFS meteorology resulted in a decrease of 4.8% in comparison with the reference simulation.



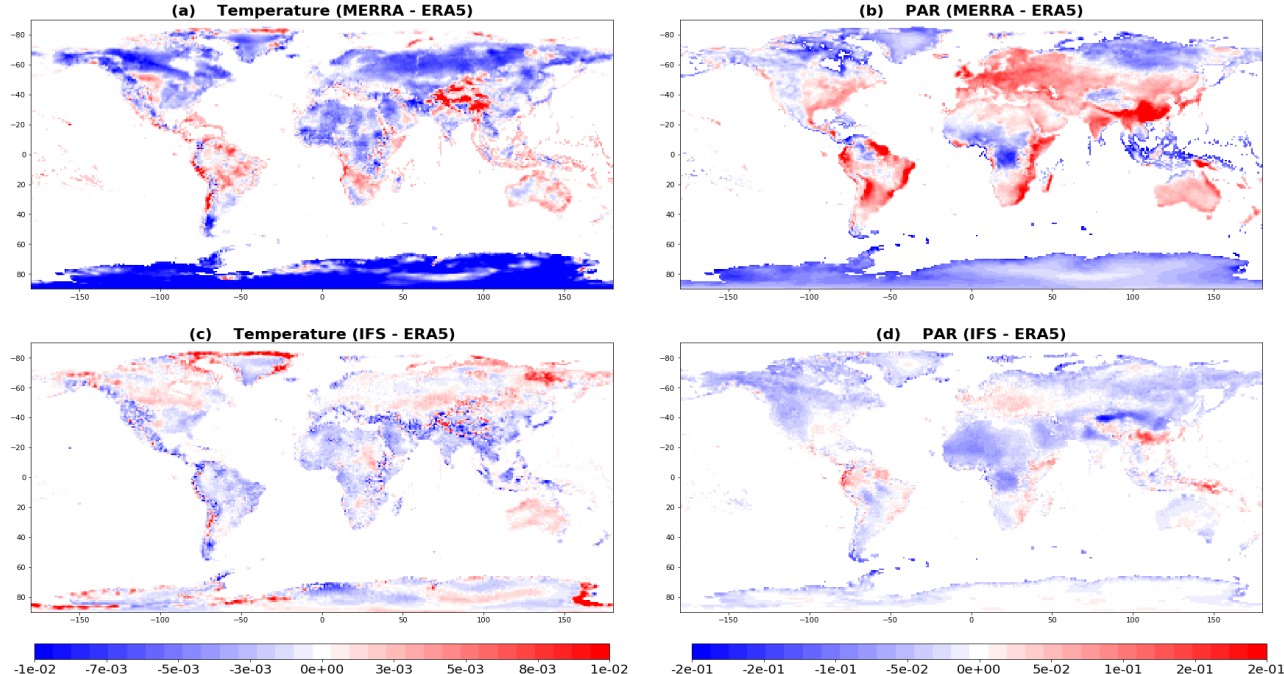

**Figure 12.** Mean temperature relative difference between MERRA and ERA5 (a), mean PAR relative difference between MERRA and ERA5 (b), mean temperature relative difference between IFS and ERA5 (c) and mean PAR relative difference between IFS and ERA5 (d). Red represents areas where the difference between temperature/PAR is positive and blue areas where the difference is negative.

## 4.2 Emission potential of isoprene

MEGANv2.1 defines two approaches to estimate biogenic fluxes, the first one is based on the use of the biogenic species emission potential maps $\epsilon_{map}$ these gridded maps are made based on a land cover including more than 2000 eco regions each with specific emission factors (Guenther et al., 2012). The compilation of these maps was done to cover the large differences in emission potential between species belonging to the same generalised PFT (e.g., temperate deciduous tree). For other PFTs, including only low isoprene emitters, the use of the PFT-specific emission factor is sufficient (e.g., boreal deciduous and needle
trees). The second approach consists of using the 16 generalised plant functional type distribution $\epsilon_{PFT}$ along with their specific emission factor (Guenther et al., 2012).

To compare between the two approaches, we have estimated global isoprene fluxes during 2019 using emission potential values $\epsilon_{PFT}$ instead of the emission potential data from the gridded maps $\epsilon_{map}$ used in the reference simulation. Figure 13 shows mean difference in isoprene emissions between the S2 simulation using $\epsilon_{PFT}$ and the reference simulation using $\epsilon_{map}$. The total
annual isoprene of simulation S2 is 390Tg, the data indicates that on a global scale, the isoprene emissions have decreased by 12%. As shown in Figure 11, this decrease is particularly pronounced in Australia (-58%) and South Africa (-25%). A notable increase is observed in Europe (+32%) and in South America (+19%), particularly in the northern Amazon.



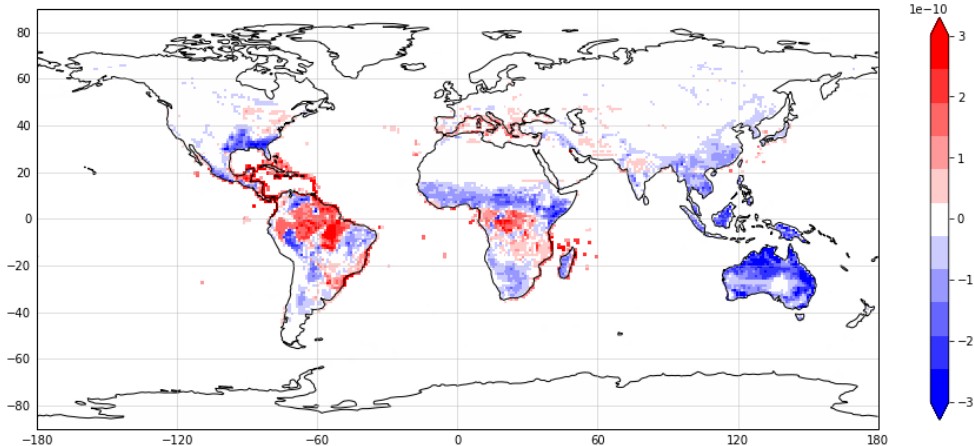

**Figure 13.** Annual mean isoprene difference (kg/m2/s) between sensitivity test S2 and reference simulation.

The results of this sensitivity test are aligned with the findings of Sindelarova et al. (2014). The MEGAN-MACC average annual isoprene emissions dropped by 14% when using the emission potential values $\epsilon_{PFT}$ instead of the emission potential
map $\epsilon_{map}$ . The decrease concerns Australia (-47%) and South Africa (-28%) and the increase concerns South America (+10%) and Europe (+18%).

The results of this sensitivity test explains partially the differences observed in section 3. CAMS-GLOB-BIOv3.0 and ALBERI inventories used $\epsilon_{PFT}$ data to estimate isoprene flux, resulting in a lower isoprene emissions compared to other datasets, as annual isoprene flux dropped by 29% and 21% respectively compared to reference simulation. Sindelarova et al. (2022) reported
a similar decrease rate in isoprene emissions estimated at 30% of CAMS-GLOB-BIOv3.0, which uses PFT-specific emission potential data and PFT distribution, compared to CAMS-GLOB-BIOv3.1, which uses isoprene emission potential gridded map.

### 4.3 Soil moisture

Prior research has investigated the association between soil moisture and isoprene emissions. Results indicate that isoprene emissions exhibit a three-phased response to drought and declining soil water. In the initial days of drought, plants tend to
retain a stable isoprene emission rate, in some instances, the emission rate may even slightly increase (Pegoraro et al., 2007). The second stage starts when soil moisture falls below a specific threshold, at which point the rate of isoprene emission begins to decrease. Extended exposure to severe drought leads to a gradual decrease in isoprene emissions, eventually, the emissions became insignificant over time (Tingey et al. (1981) - Pegoraro et al. (2004b) - Wang et al. (2021) - Wang et al. (2022) - Trimmel et al. (2023)).
The response of isoprene emission to drought is simulated in MEGAN indirectly by the MEGAN canopy environment model by incorporating the leaf temperature estimate, which is affected by soil moisture. MEGAN also includes a $\gamma_{SM}$ factor which





directly simulates the response of isoprene emission to drought. This factor is derived from soil moisture parameterization experiments conducted by Pegoraro et al. (2004a). The $\gamma_{SM}$ is defined as follows:

$$\gamma_{SM} = 1 \quad \theta > \theta_1 \qquad\qquad \gamma_{SM} = \frac{(\theta - \theta_w)}{\Delta\theta_1} \quad \theta_w < \theta < \theta_1 \qquad\qquad \gamma_{SM} = 0 \quad \theta < \theta_w \tag{8}$$

where $\theta$ is soil moisture (volumetric water content, m3 m-3), $\theta_w$ (m3 m-3) is wilting point (the soil moisture level below which plants cannot absorb water from soil) and $\Delta\theta_1$ (=0.04) is an empirical parameter and $\theta_1 = \theta_w + \Delta\theta_1$ (Guenther et al., 2012).

The third sensitivity test (S3) was conducted to examine the effect of soil moisture on isoprene emissions. To estimate $\gamma_{SM}$, MEGAN uses wilting point data calculated in SURFEX from the sand and clay covers given as input to the coupled model following Clapp and Hornberger (1978) and Lepistö et al. (1988) approaches. The sand and clay data are extracted from HWSD

(The Harmonised World Soil Database), which is a global soil database developed by the FAO (Food and Agriculture Organisation of the United Nations) in collaboration with IIASA (International Institute for Applied Systems Analysis) in order to provide information on the physical and chemical properties of soils across the world.

In order to accurately estimate soil moisture, a four-year spin-up period was required to stabilise the soil water content with the ISBA force-restore 2-L scheme. This approach is used to simulate the exchange of energy and water between the surface

and the atmosphere and is based on the balance between the forces that drive the exchange of energy and water (radiation, temperature, and precipitation) and the restoring forces that return the system to equilibrium (evaporation, transpiration, and runoff) (Boone et al., 1999) (Hu and Islam, 1995). The wilting point and soil water content are calculated at different soil layers, depending on the ISBA scheme model used. In ISBA force-restore 2-L scheme, the soil is represented with two layers. In the present study, to evaluate soil moisture impact on isoprene emissions, we have used soil moisture and wilting point data

from the second layer, as it most accurately represents the root depth of the vegetation.

The integration of the soil moisture algorithms led to a total isoprene emissions of 273Tg, with a global decrease of 38% compared to the reference simulation. Figure 14, 15 and 16 show the mean annual isoprene difference between S3 simulation and reference simulation, the spatial distribution of average $\gamma_{SM}$ over 2019 and the mean annual soil liquid water content estimated at the second ISBA-2L layer as well as the relative wilting point data used in the S3 simulation respectively. The

decrease concerns mainly arid and semi-arid regions, the largest decrease can be observed in Australia (-89%), followed by North Africa and Middle East (-82%), South Africa (-67%) and East Africa (-38%). In South America, emissions are lower by 23% and the decrease is mainly located in Brazil.

Previous studies have studied the impact of soil moisture on isoprene emissions and have reported varying decrease rates. Guenther et al. (2006) obtained the lowest decrease rate of 7%, Müller et al. (2008) found a decrease rate of 21% and Sinde-

larova et al. (2014) reported the highest decrease rate of 50%. The discrepancies in the reported values of isoprene decrease rate can be attributed to the use of different soil moisture and wilting point data. The latter, is a critical parameter, as it defines the limit below which, the soil moisture activity factor is set to 0, consequently, Guenther et al. (2012) stressed the importance of using consistent wilting point data with the soil moisture input. In this context, SURFEX-MEGAN model enhances the



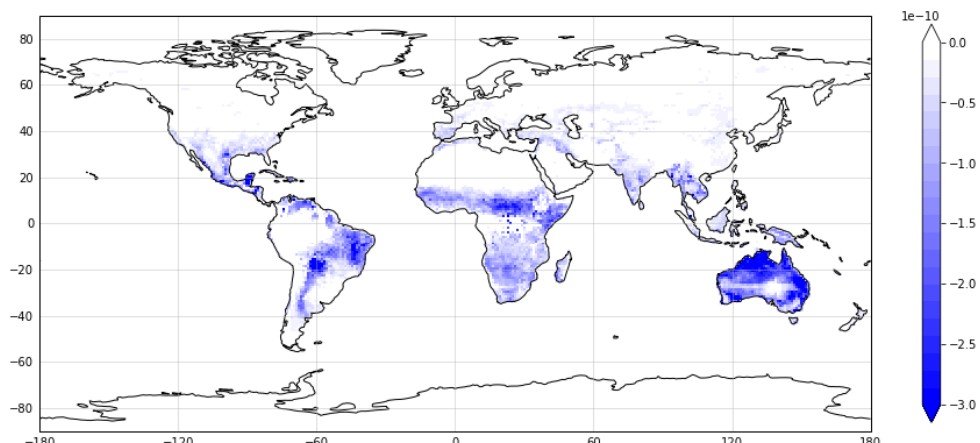

**Figure 14.** Annual mean isoprene difference (kg/m2/s) between sensitivity test S3 and reference simulation.

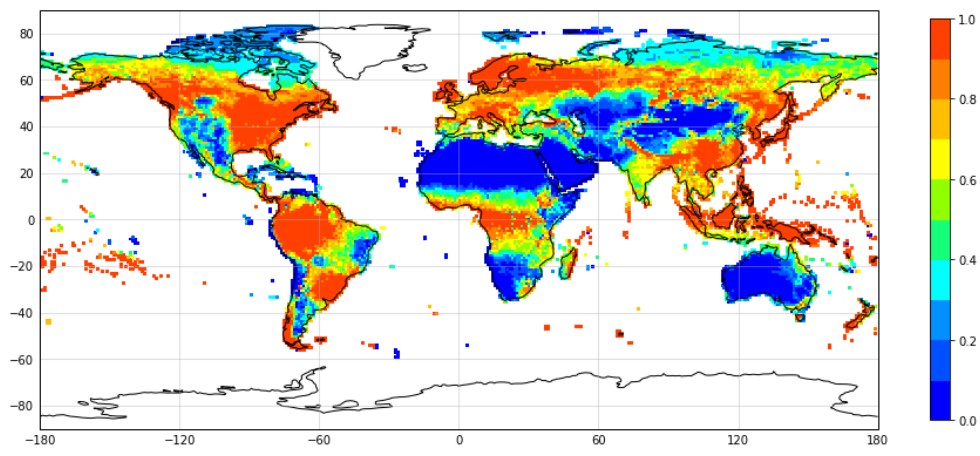

**Figure 15.** Spatial distribution of the mean annual soil moisture dependence factor $\gamma_{SM}$ in S3 simulation.

precision of $\gamma_{SM}$ calculation by using vegetation type-dependent soil moisture at a given layer and wilting point data at the
same soil layer.

## 5    Conclusions

The presented paper describes the implementation of the biogenic model MEGANv2.1 (Guenther et al., 2012) in the surface
model SURFEX (Le Moigne, 2018). The aim of this coupling is to improve the accuracy of vegetation type-specific parame-
ters for MEGAN2.1 by leveraging the detailed canopy environment model built into SURFEX. This improved accuracy, should
lead to better estimates of BVOCs.



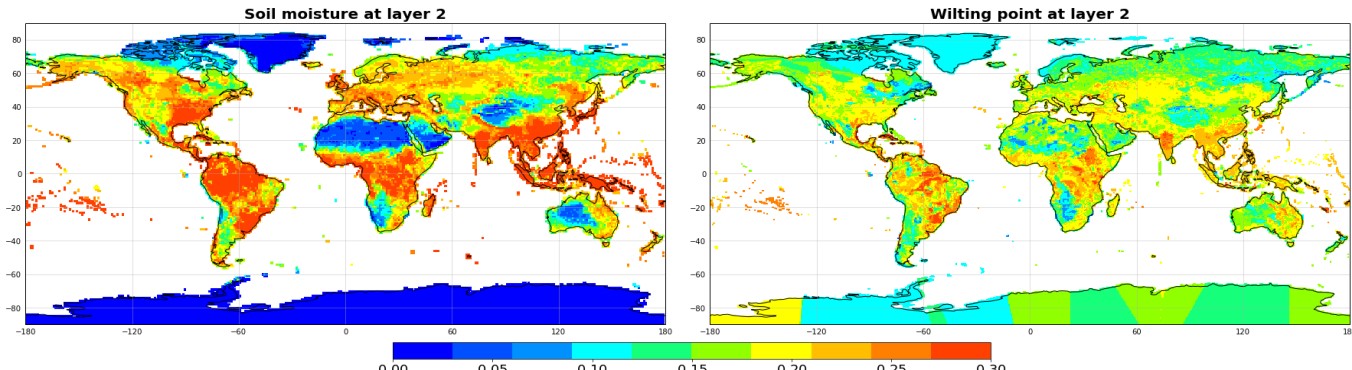

**Figure 16.** Annual average soil liquid water content (m3/m3) (left) and wilting point data (m3/m3) (right) of the ISBA-2L second layer.

The coupling validation was done by running a global simulation (1°, hourly) in 2019 using ERA5 meteorological data inputs. The total annual isoprene is estimated to be 443Tg. The SURFEX-MEGAN total annual isoprene is within the range of isoprene estimates reported in previous studies. To validate the coupled model, the 2019 isoprene simulation results were compared to isoprene estimates of 3 previous published studies. A spatial and temporal analysis were conducted to compare

the different results. The SURFEX-MEGAN emission estimates were shown to have a comparable spatial distribution to the other inventories, especially to the ones using similar setup (e.g., meteorology, emission potential data). As for the monthly variation of isoprene emissions, SURFEX-MEGAN follows the same temporal pattern as some of the inventories, the shift in the annual isoprene cycle was explained by the difference in the contribution of the emitting regions to the global isoprene for each inventory.

A list of sensitivity tests was performed to investigate the impact of key MEGAN variables on isoprene emissions. To highlight the difference between the coupled SURFEX-MEGAN model and other MEGAN-based models, the results of the sensitivity tests were compared with the findings of other studies. The use of different meteorological forcings resulted in isoprene estimates varying up to +/- 5% of the reference run results, with Australia, South America and Africa being the most affected regions. The use of different input of emission potential data led to a decrease of 14% globally. The activation of the soil

moisture parametrization was shown to have the greatest impact on isoprene emissions. On a global scale, the emission have decreased by 38%, the largest decrease was observed in Australia (-89%) and in Africa. The decrease rate related to the activation of the soil moisture activity factor varies across different studies, which has been attributed to inconsistencies in the soil moisture and wilting point data employed. The SURFEX-MEGAN model offers an advantage in this regard, as it can compute the wilting point and soil moisture at the same soil layer for different vegetation types, leading to a more precise estimation of

the gamma soil moisture. This high sensitivity to soil moisture emphasises the importance of conducting further studies in this area in order to reduce uncertainties, in particular by refining the estimation of the empirical parameter $\Delta\theta_1$.

The potential perspectives to be explored from this study concern the assessment of biogenic emissions in future climates as BVOCs are expected to undergo significant changes resulting from the alteration of biogenic emission climate drivers. This



assessment is particularly relevant to air quality forecasting in the context of ongoing global warming and predicted future climate change. In this respect, the particularity of SURFEX lies in its ability to be used in offline mode as it can be forced with future climate meteorology. SURFEX also includes a biomass evolution sub-model, allowing the evolution of vegetation density (leaf area index) as a function of changing meteorological and environmental variables. This feature would be of particular use for predicting biogenic emissions under future climate scenarios whereby the evolution in vegetation could be simulated in SURFEX using the dynamic LAI vegetation scheme.

*Code availability.* The SURFEX code is available at: https://www.umr-cnrm.fr/surfex/spip.php?rubrique8. The MEGAN code is available at: https://bai.ess.uci.edu/megan/data-and-code/megan21. The code for the coupled model SURFEX-MEGAN can be made available upon request from Safae Oumami (safae.oumami@meteo.fr).

*Author contributions.* PT contributed to the implementation of MEGAN in MesoNH, to editing and revision of the paper. SO implemented and developed the updates of the online SURFEX-MEGAN coupling, performed the simulations and complementary analyses, and drafted the paper. VG and JA contributed to the design of the simulations, provision of the data, analysis and interpretation of the results, and editing and revision of the paper. PH contributed to the editing and revision of the paper.

*Competing interests.* The authors declare that they have no competing interests.

*Acknowledgements.* We thank Alex Guenther and the University of California, Irvine (UCI) for providing the MEGANv2.1 code and the necessary data. We thank the SURFEX team at CNRM for their invaluable assistance in facilitating the online coupling of SURFEX-MEGAN, as well as for the provided access to the SURFEXv8.1 code and physiographic fields data. The ERA5 and IFS meteorological data were provided by the European Centre for Medium-Range Weather Forecasts (ECMWF). The MERRA meteorological data was provided by the NASA's Global Modelling and Assimilation Office (GMAO). The MEGAN-MACC and CAMS-GLOB-BIO isoprene emissions were extracted from the Emissions of atmospheric Compounds and Compilation of Ancillary Data (ECCAD) database. The ALBERI isoprene dataset was made available by the Tropospheric Modelling team of the Royal Belgian Institute for Space Aeronomy (BIRA-IASB).



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
