# Peer review of "Evaluation of isoprene emissions from the coupled model SURFEX-MEGANv2.1"

_EGUsphere, 2023_

## Author Response (AR1)

We would like to thank the referee for their very helpful and constructive feedback. They have identified some areas where we are able to greatly improve the review.

We have responded to the referee's comments in blue font below.

**Anonymous Referee #1**

The authors provide a thorough overview of the implementation of the MEGAN module into the SURFEX land model, and compare the model results in terms of isoprene emissions against reported inventories from the literature. The authors obtain mostly similar emission totals and distributions, compared to other MEGAN-based emission estimates, and clearly discuss the impact of model assumptions and input datasets which drive the parameterization. The impact of changes in driving meteorology is relatively modest, and mostly explained by the PPFD. Changes in emission potential can have a larger impact, while the introduction of a sensitivity to drought, through the actual soil moisture relative to the wilting point, were shown to lead to very large differences in modeled emissions. Based on this information the authors rightly address this issue that further studies are necessary to reduce the uncertainties to soil moisture assumptions.

We thank referee 1 for the concise summary of the paper.

This raises the question whether the authors have any thoughts on how they intend to validate the assumed BVOC emissions. It would be useful if they could spend some words on this. It also relates to a concern I have whether there is evidence if some of the emission estimates (esp. the MEGAN-MACC) can be disregarded from this study, as they may appear beyond the range of reasonable values.

While we think that the validation of the modelled BVOC emissions against flux measurements is of great value, it is not the primary focus of our paper. The paper seeks to validate the implementation of MEGAN in SURFEX by comparing the coupled model SURFEX-MEGAN isoprene emissions with other MEGAN-based isoprene inventories. Furthermore, the validation of global isoprene emissions is of extreme difficulty due the lack of isoprene observations. The sensitivity to the emission-driving variables and the uncertainties related to the MEGAN model was thoroughly discussed in other papers aiming to compare the MEGAN model estimations to local isoprene flux measurements ((Sindelarova et al.2014) - (Situ et al.2014) – (Kota et al.2015) - (Seco et al.2022)).

References:

Sindelarova, K., Granier, C., Bouarar, I., Guenther, A., Tilmes, S., Stavrakou, T., ... & Knorr, W. (2014). Global data set of biogenic VOC emissions calculated by the MEGAN model over the last 30 years. *Atmospheric Chemistry and Physics*, *14*(17), 9317-9341.

Situ, S., Wang, X., Guenther, A., Zhang, Y., Wang, X., Huang, M., ... & Xiong, Z. (2014). Uncertainties of isoprene emissions in the MEGAN model estimated for a coniferous and broad-leaved mixed forest in Southern China. *Atmospheric environment*, *98*, 105-110.

Seco, R., Holst, T., Davie-Martin, C. L., Simin, T., Guenther, A., Pirk, N., ... & Rinnan, R. (2022). Strong isoprene emission response to temperature in tundra vegetation. *Proceedings of the National Academy of Sciences*, *119*(38), e2118014119.

Kota, S. H., Schade, G., Estes, M., Boyer, D., & Ying, Q. (2015). Evaluation of MEGAN predicted biogenic isoprene emissions at urban locations in Southeast Texas. *Atmospheric Environment*, *110*, 54-64.

We have added this paragraph to clarify the aim of this paper: "The present paper aims to validate the coupling of the land model SURFEX and the biogenic model MEGAN by comparing the model results to

other MEGAN-based inventories. This validation does not include a comparison to real world observations as the MEGAN model was thoroughly discussed in other papers aiming to validate the MEGAN model estimations to local isoprene flux measurements ((Sindelarova et al.2014) - (Situ et al.2014) – (Kota et al.2015) - (Seco et al.2022))."

Minor comments:

line 8: "more precise": more precise than what?

"This scheme provides to MEGAN vegetation-dependent parameters allowing a more precise estimation of biogenic fluxes (e.g., leaf area index, soil moisture, wilting point data)."

Will be replaced with:

"This scheme provides to MEGAN vegetation-dependent parameters such as leaf area index and soil moisture. This approach enables a more accurate estimation of biogenic fluxes compared to the stand-alone MEGAN model, which relies on average input values for all vegetation types.".

line 42" emission of ozone" -> "formation of ozone"?

Changed as suggested.

Figure 1 / line 95 / line 96 / line 97: Please align the definition of different tiles, esp choose between 'Town' / Urban area / city , but not these three different names.

We have used "Town".

line 183 "its corresponding defined" please check wording

"Each vegetation type from ECOCLIMAP-II was mapped to its corresponding defined in CLM4"

Will be replaced with:

"Each vegetation type from ECOCLIMAP-II was mapped to its corresponding type defined in CLM4."

line 195 "is similar for most vegetation types": Actually, apart from shrubs I also see relatively large discrepancies for grassland and needle-leaf trees - but maybe it's a colorscale issue (and/or not so relevant in eventual emissions.) - can't the authors comment?

Indeed, there is a disparity in needleleaf tree coverage between ECOCLIMAP and CLM4, however, the impact on global isoprene emissions is expected to be minor. This is attributed to the fact that this specific plant functional type represents only 1.4% of the total annual emitted isoprene. (Guenther et al.2012).

We have added

"For other vegetation types (e.g., needeleleaf trees), the difference in vegetation density between ECOCLIMAP and CLM4 is expected to have a small impact on isoprene emissions, as this specific PFT represents only 1.4% of the total annual emitted isoprene (Guenther et al.2012)."

References:

Guenther, A. B., Jiang, X., Heald, C. L., Sakulyanontvittaya, T., Duhl, T. A., Emmons, L. K., & Wang, X. (2012). The Model of Emissions of Gases and Aerosols from Nature version 2.1 (MEGAN2. 1): an extended and updated framework for modeling biogenic emissions. *Geoscientific Model Development*, *5*(6), 1471-1492.

line 200: "LAI value of the past 10 days" change to "LAI value of 10 days in the past"?

Changed as suggested.

Figure 3 shows that also the T24 and PPFD24 (previous day mean temperature and PPFD) are required as input to the ISBA-MEGAN processor. Out of curiosity, how are these parameters computed? Simply keeping track of temporal variations in PPFD and Temperature over all (24) hours for the last day, or is there a more intricate procedure for this?

T24 and PPFD24 are calculated by storing the 24h historical values of temperature and light, the averaged values are updated at 00:00 of each day.

line 217 "simulation of isoprene emissions"

Changed as suggested.

line 224: The authors write: "As there are no available inputs for surface incident diffuse shortwave radiation ... a value of 0Wm-2 is assigned". Maybe I do not understand this well, but wouldn't it make more sense to assign a higher default value than 0 Wm-2 for this quantity?

We agree that using a higher value for shortwave diffused radiation would be more reasonable. The decision to assign a value of 0 to the diffused component of radiation was driven by the challenge of identifying a default value that accurately represents the spatial and temporal variations of diffused shortwave radiation.

line 275: "PAR calculated from incoming shortwave radiation" from which product is this PAR here derived? From ERA5 / ERA-Interim? please clarify

This paragraph shows the impact of using different PAR inputs on isoprene emissions. Sindelarova et al. (2014) performed a sensitivity study using two approaches. The first derives PAR from incoming shortwave radiation (SR) obtained from the MERRA reanalysis data (PAR = 0.5 * SR) and the second uses the PAR provided on hourly basis by the MERRA Land model.

"Sindelarova et al. (2014) found that using PAR calculated from incoming shortwave radiation instead of PAR from the MERRA reanalysis led to a 17.5% increase in total isoprene emissions. »

Will be replaced with

"Sindelarova et al. (2014) found that using PAR derived from the MERRA incoming shortwave radiation, instead of PAR provided by the MERRA land model led to a 17.5% increase in total isoprene emissions."

References:

Sindelarova, K., Granier, C., Bouarar, I., Guenther, A., Tilmes, S., Stavrakou, T., ... & Knorr, W. (2014). Global data set of biogenic VOC emissions calculated by the MEGAN model over the last 30 years. *Atmospheric Chemistry and Physics*, *14*(17), 9317-9341.

Figure 6: It would help if the authors re-organize the grouping such that one panel is given for each region, with bars for each emission product, to more easily intercompare the various estimates per region and month.

Changed as suggested (see Figure below).

[Figure]

Figure 5: Contribution of zonal regions to monthly isoprene in CAMS-GLOB-BIOv1.2, CAMS-GLOB-BIOv3.0, CAMS-GLOB-BIOv3.1, MEGAN-MACC, ALBERI and SURFEX-MEGAN reference simulation in 2019 (2018 for CAMS-GLOB-BIOv1.2 and ALBERI).

**Anonymous Referee #2**

Comments:

As often mentioned in the manuscript, the goal of this work is to force the coupled model using climate change scenarios and assess the impact of climate change on the biogenic emissions. However, only a single year (2019) is presented in this paper. Unless you include long SURFEX-MEGAN simulation results in a revised version, it should be avoided to create more expectations than the manuscript can address.

This point was discussed in the paper to stress the motivation behind the work of coupling the land model SURFEX and the biogenic model MEGAN. We agree that this might create more expectation than the manuscript addresses. In a revised version, this point will only be mentioned in the introduction of the paper and briefly as a perspective in the conclusions section.

We have replaced "Our scientific aim was to derive a method for estimating BVOC emissions for present and future climate scenarios that would be capable of considering both atmosphere and land surface processes as well as land-atmosphere interactions that impact vegetation."

with "Our scientific aim was to derive a method for estimating BVOC emissions that would be capable of considering both atmosphere and land surface processes as well as land-atmosphere interactions that impact vegetation."

We have replaced "First, SURFEX can be used in offline mode (i.e. using an external meteorological forcing file), this option enables simulations to be performed in present and future climates."

With "First, SURFEX can be used in offline mode (i.e. using an external meteorological forcing file)."

We have replaced "Additionally, this scheme can simulate LAI, which varies in parallel with numerous environmental and meteorological variables. Based on this dynamic LAI, the coupled model can assess and predict the impact of climate change on the biosphere."

With "Additionally, this scheme can simulate LAI, which varies in parallel with numerous environmental and meteorological variables. Based on this dynamic LAI, the coupled model can estimate biogenic emissions interactively with leaf biomass."

You mention that the vegetation-type specific treatment in SURFEX improves the accuracy of isoprene estimates. Please provide evidence to prove this. Data from isoprene flux measurement campaigns (Seco et al. 2022, Emmerson et al., 2020; and many others) should be used to assess the SURFEX-MEGAN emissions. In a revised version, a detailed evaluation of the model results against flux measurements should be included.

As stated in the paper, SURFEX includes routines that allow for the treatment of each vegetation type separately. This means that the estimation of several vegetation-related parameters (e.g., soil moisture, soil temperature, leaf area index) will be carried out for each vegetation type independently following parametrization governing the physics of each vegetation type. Since the estimation of isoprene flux in MEGAN relies on these parameters, our assumption is that using parameters for each vegetation type, as opposed to using a single averaged value for all vegetation types, results in a more accurate representation of input parameters. This should, in turn, lead to a more accurate estimation of isoprene following the MEGAN parametrization. More in-depth justification of this reasoning is provided below.

In fact, the ISBA scheme effectively captures the unique physics of different plant functional types (PFTs), such as grasslands and dense canopy forests. This differentiation is critical because each PFT is defined by specific parameters, including root depth. Root depth is a key factor in accurately

representing how each vegetation type accesses soil moisture, which in turn influences the soil moisture activity factor estimated within SURFEX. While certain ISBA variables like soil moisture and Leaf Area Index (LAI) have been validated through field measurements, the model estimations at the patch level were not validated due to measurement limitations. Nevertheless, ISBA has demonstrated reliable performance in estimating biomass, leaf area index, evapotranspiration, soil carbon, etc compared to other land surface models (Friedlingstein et al. 2022).

Therefore, when it is said that SURFEX improves isoprene estimation, it specifically refers to the technical capability of SURFEX to provide more accurate input parameters compared to the standalone MEGAN version, which uses an average value for all vegetation types.

While we think that it is important to validate the MEGAN estimations against field measurements, it is not the primary focus of this paper. This manuscript seeks only to validate the implementation of MEGAN in SURFEX by comparing the coupled model SURFEX-MEGAN isoprene emissions with other MEGAN-based isoprene inventories. To also include a validation of the coupled model SURFEX-MEGAN isoprene estimates against field measurements for each PFT would, we believe, significantly increase the size of an already long manuscript.

We have added this paragraph to clarify the aim of this paper: "The present paper aims to validate the coupling of the land model SURFEX and the biogenic model MEGAN by comparing the model results to other MEGAN-based inventories. This validation does not include a comparison to real world observations as the MEGAN model was thoroughly discussed in other papers aiming to validate the MEGAN model estimations to local isoprene flux measurements ((Sindelarova et al.2014) - (Situ et al.2014) – (Kota et al.2015) - (Seco et al.2022))."

References:

Friedlingstein, P., Jones, M. W., O'sullivan, M., Andrew, R. M., Bakker, D. C., Hauck, J., ... & Zeng, J. (2022). Global carbon budget 2021. *Earth System Science Data*, *14*(4), 1917-2005.

- The differences between the reference run and the sensitivity run using MERRA reanalysis are not adequately explained. To ease the discussion, could you include comparisons of the monthly temperature from ERA5 and MERRA for the regions of Figure 8? Could you quantify the differences between the two datasets for temperature and solar radiation, which are the main driving factors of the isoprene emissions? Where are those differences more significant and what is the induced uncertainty due to the meteorology?

We have added in this section the following paragraph to try to illustrate the differences between the ERA5 and MERRA temperature and PAR and the impact on isoprene estimation:

"A regional analysis was also conducted to quantify the impact of using different meteorological datasets on isoprene estimates. Figure 13 displays monthly isoprene emissions of the reference and S1-MERRA simulations across the globe regions shown in Figure 8. Isoprene flux absolute difference is mostly pronounced in Australia, South America and Southern Africa, where S1-MERRA isoprene estimates are higher than the reference simulation. In South America, Southern Africa and Australia, S1-MERRA monthly isoprene emissions are higher than the reference simulation by a range of 2%-10%, 1%-11% and 6%-15%.

In these regions, although the temperature difference between MERRA and ERA5 is small (less than 0.5°), the photosynthetic active radiation PAR difference is significant. In these regions, PAR variations range between -1-9w/m², -2-8w/m2 and -1-8w/m², respectively. Consequently, the main factor driving

monthly variations in isoprene emissions between the reference simulation and the S1-MERRA simulation is PAR."

- It is not clear what parameterization is used for the activity factor accounting for the isoprene inhibition due to enhanced $CO_2$ levels (Equ. 2). The impact of $CO_2$ levels on isoprene fluxes is highly uncertain, as past work reported contradictory findings (Sun et al., 2013; Tai et al., 2013, Bauwens et al., 2018). I urge the authors to discussion this aspect and quantify the impact of the $CO_2$ inhibition on the total annual emission.

In our SURFEX-MEGAN model, we incorporate the CO2 inhibition parametrization from Heald et al. (2009). We chose not to focus on this aspect in our paper because the inhibitory effect of CO2 on isoprene emission becomes significant only when atmospheric CO2 levels exceed 400ppmv

[Figure]

Figure 13: Monthly variation of isoprene flux from reference and S1-MERRA simulations of the globe regions defined in Figure 8 in kg/m²/s.

substantially, as indicated by Sindelarova et al.(2014). This consideration is primarily relevant for future scenarios where CO2 concentrations are expected to rise.

We have added this sentence in the Model setup section to explain the choice of setting this parameter to 1:

"Unless otherwise stated, in all coupled model simulations the estimation of isoprene flux was done based on isoprene potential map and the effect of soil moisture deficit and CO2 on BVOC emissions was not taken into account (the γsm andγCO2 factors were assigned to 1). This choice allows a better comparison with other emission inventories. Additionally, the impact of the CO2 inhibition factor becomes relevant only when CO2 atmospheric concentrations exceed significantly 400ppmv (Sindelarova et al. (2014))."

References:

Heald, C. L., Wilkinson, M. J., Monson, R. K., Alo, C. A., Wang, G., & Guenther, A. (2009). Response of isoprene emission to ambient CO2 changes and implications for global budgets. *Global Change Biology*, *15*(5), 1127-1140.

Sindelarova, K., Granier, C., Bouarar, I., Guenther, A., Tilmes, S., Stavrakou, T., ... & Knorr, W. (2014). Global data set of biogenic VOC emissions calculated by the MEGAN model over the last 30 years. *Atmospheric Chemistry and Physics*, *14*(17), 9317-9341.

- This study does not account for recent efforts to improve the soil moisture activity factor parameterization in MEGAN (Jiang et al., 2018; Wang et al., 2022; Opacka et al. 2022). A discussion is needed and an estimation of the isoprene fluxes using (at least) one different parameterization should be added in the manuscript.

We agree with the referee on this point. However, there are some limitations to conducting more analysis on soil moisture activity factor. First, As mentioned in Opacka et al.2022, the impact of soil moisture can be refined in MEGANv2.1 by improving the estimation of the empirical parameter $\Delta\theta$. In earlier studies this parameter was fixed to 0.06 but was re-adapted to a new value of 0.04. This new value was chosen so that the emissions are shut off only in extreme drought events. Therefore, this study focused on improving the estimation of the parameter $\Delta\theta$ based on isoprene flux measurements at the Missouri Ozarks AmeriFlux (MOFLUX) site. As mentioned in this paper, the adjustment of $\Delta\theta$ is influenced by soil moisture $\theta$ data and the soil depth of reference, thus, the $\Delta\theta$ adjusted value found in this paper can not be applied directly to our model. We believe that conducting a thorough study on the adjustment of this empirical parameter is not straightforward and it would require a consequent work. Therefore, this particular point will be addressed in a future paper focusing only on the optimization of soil moisture activity factor in the coupled model SURFEX-MEGAN.

Other approaches exist to improve the estimation of soil moisture impact on isoprene emissions. These approaches are based on the new soil moisture parametrization used in MEGANv3. As discussed in Jiang et al.2018, this new approach improves the estimation of isoprene in drought and non-drought events. The estimation of the soil moisture activity factor used in MEGANv3 is based on two parameters $V_{cmax}$ and $\beta_t$. The former represents the maximum rate of carboxylation by the photosynthetic enzyme Rubisco and the latter the soil water stress function. This approach was tested in MEGANv2.1 embedded in the Community Land Model CLM4, which has detailed biogeophysical and hydrological cycles, and biogeochemical components, and can estimate carbon, water, and energy fluxes. As opposed to CLM4, the current version of SURFEX, does not include a direct estimation of $V_{cmax}$ and $\beta_t$. The inclusion of these parameters would require considerable work. Therefore, we agree with the referee on the importance of exploring alternative approaches for the soil moisture activity factor, however, considering the potential time investment, we propose conducting separate tests for these approaches in a forthcoming paper.

We have added the following paragraph to discuss the limitations of the MEGANv2.1 soil moisture parametrization:

"
Several limitations associated with the use of this MEGANv2.1 soil moisture parametrization have been identified. Primarily, the parametrization exhibits a significant dependency on the wilting point data, which can show inconsistency with the soil moisture data used. Furthermore, it has been shown that this parametrization substantially reduces isoprene emissions, even under moderate drought conditions, thereby indicating a potential oversensitivity to drought stress (Wang et al.(2022)). The

introduction of the new MEGAN3 soil moisture factor addresses these shortcomings by providing robust performance under both moderate and severe drought conditions (Jiang et al.(2018)). This enhancement in soil moisture representation is anticipated to be integrated into the forthcoming version of the SURFEX-MEGAN coupled model, thereby offering a more accurate and reliable prediction of isoprene emissions under varying soil moisture scenarios."

References:

Opacka, B., Müller, J. F., Stavrakou, T., Miralles, D. G., Koppa, A., Pagán, B. R., ... & Guenther, A. B. (2022). Impact of drought on isoprene fluxes assessed using field data, satellite-based GLEAM soil moisture and HCHO observations from OMI. *Remote Sensing*, *14*(9), 2022.

Wang, H., Lu, X., Seco, R., Stavrakou, T., Karl, T., Jiang, X., ... & Guenther, A. B. (2022). Modeling isoprene emission response to drought and heatwaves within MEGAN using evapotranspiration data and by coupling with the community land model. *Journal of Advances in Modeling Earth Systems*, *14*(12), e2022MS003174.

Jiang, X., Guenther, A., Potosnak, M., Geron, C., Seco, R., Karl, T., ... & Pallardy, S. (2018). Isoprene emission response to drought and the impact on global atmospheric chemistry. *Atmospheric Environment*, *183*, 69-83.

- About Figures: (i) The readability of Fig 2 is quite low. Could you change the scale? (ii) Not clear where Fig 8 is used. The figure seems to be at lower resolution. Can you improve the resolution?

We will change the scale of Figure 2.

Figure 8 has been included to provide a visual representation of the geographical regions utilized in the analysis. However, we intend to substitute this figure with another one with a higher resolution.

- l 74-75, "the coupling between SURFEX and MEGAN can create a feedback loop that takes into account both the impact on climate on vegetation and the impact of vegetation on climate'. Poorly worded sentence. Furthermore, these impacts are not presented in the manuscript. Could you include a tentative quantification of these impacts?

We have amended this sentence to clarify the feedback loop between vegetation and climate:

"The impact of vegetation on climate can also be investigated through the Earth System Model CNRM-ESM2-1 (Séférian et al.2019), which includes the land surface model SURFEX. This effect originates from the BVOCs-induced impact on aerosols and other greenhouse gases concentration (ozone and methane), which can alter the Earth's radiative balance (Unger et al.2014(a) – Unger et al.2014(b) – Sporre et al.2019)."

References:

Séférian, R., Nabat, P., Michou, M., Saint-Martin, D., Voldoire, A., Colin, J., ... & Madec, G. (2019). Evaluation of CNRM Earth System Model, CNRM-ESM2-1: Role of Earth system processes in present-day and future climate. *Journal of Advances in Modeling Earth Systems*, *11*(12), 4182-4227.

Unger, N. (2014). Human land-use-driven reduction of forest volatiles cools global climate. *Nature Climate Change*, *4*(10), 907-910.

Unger, N. (2014). On the role of plant volatiles in anthropogenic global climate change. *Geophysical Research Letters*, *41*(23), 8563-8569.

Sporre, M. K., Blichner, S. M., Karset, I. H., Makkonen, R., & Berntsen, T. K. (2019). BVOC–aerosol–climate feedbacks investigated using NorESM. *Atmospheric Chemistry and Physics*, *19*(7), 4763-4782.

- l 291: 'In low isoprene emission regions, such as North America'. This is not correct. Southeast US is among the strongest isoprene emitting regions in the world (Wiedinmyer et al., 2005; Kaiser et al. 2017 and others).

Will be replaced with "In other globe regions, such as North America, Europe, and North Asia, isoprene emissions from SURFEX-MEGAN are particularly higher when compared to other isoprene inventories."

- l 314: 'by a very low activity', replace by 'a very low flux'

Changed as suggested.

- l 323: 'These regions are active', replace by 'These regions are emitting'

Changed as suggested.

- l 330: 'As stated in (Guenther et al., 2012), replace by 'As reported in Guenther et al. (2012)'. Here and throughout the manuscript, check your citations.

Changed as suggested.

- Table 4: can you add a column with the global total emission for each of the setups? Similarly in Figure 4: add the global emission estimate inside the subplots. In Table 4, the description 'use of isoprene emission potential' is vague. Be more specific.

Global total emission will be added in Table 4 and Figure 4. In Table 4 'use of isoprene emission potential' will be replaced with 'use of PFT-specific isoprene emission potential data $e_{PFT}$'.

- l 346: correct problems with spaces

Problem solved.

- Fig 13: Are the red dots over islands due to very low or zero $e_{map}$ values over those regions? I recommand to add a global map of isoprene flux based on the $e_{PFT}$ (with the global emission provided inset).

Red dots over islands are due zero $e_{map}$ values. S2 global isoprene flux map will be added.

We have added

"The red dots over islands shown in this Figure are due to the fact that the isoprene emission factor from the input emission potential map is equal to zero in these areas."

- l 385: 'has been observed', replace by 'has been calculated' or 'has been found'

Changed as suggested.

- l 392: 'was done to cover', replace by 'accounts for'

Changed as suggested.

- l 425: correct the exponents

Corrected.

- l 421: '273Tg'. Add a space '273 Tg' here and throughout the manuscript.

Corrected.

- code availability: provide links for the datasets presented in the paper

We have provided a link for the datasets and for the model code.